# A foundational large language model for edible plant genomes
Javier Mendoza-Revilla[1,3], Evan Trop[1,3], Liam Gonzalez[1], Maša Roller[1], Hugo Dalla-Torre[1],
Bernardo P. de Almeida[1], Guillaume Richard[1], Jonathan Caton[2], Nicolas Lopez Carranza[1],
Marcin Skwark [1], Alex Laterre[1], Karim Beguir[1], Thomas Pierrot [1,4] ✉ & Marie Lopez [1,4] ✉

Significant progress has been made in the field of plant genomics, as demonstrated by the increased use of high-throughput methodologies that enable the characterization of multiple genome-wide molecular phenotypes. These findings have provided valuable insights into plant traits and their underlying genetic mechanisms, particularly in model plant species. Nonetheless, effectively leveraging them to make accurate predictions represents a critical step in crop genomic improvement. We present AgroNT, a foundational large language model trained on genomes from 48 plant species with a predominant focus on crop species. We show that AgroNT can obtain state-of-the-art predictions for regulatory annotations, promoter/terminator strength, tissue-specific gene expression, and prioritize functional variants. We conduct a large-scale in silico saturation mutagenesis analysis on cassava to evaluate the regulatory impact of over 10 million mutations and provide their predicted effects as a resource for variant characterization. Finally, we propose the use of the diverse datasets compiled here as the Plants Genomic Benchmark (PGB), providing a comprehensive benchmark for deep learning-based methods in plant genomic research. The pre-trained AgroNT model is publicly available on HuggingFace at https://huggingface.co/InstaDeepAI/agro-nucleotide-transformer-1b for future research purposes.

The advent of high-throughput next-generation sequencing has led to a vast increase in available genomic data in the field of plant sciences. Since the completion of the genome sequence of the model plant *Arabidopsis thaliana* over 20 years ago[1], more than 200 plant species' genome sequences have been published to date[2]. However, the generation of a species' assembly is merely the initial step in the process of understanding the genome. Extensive additional experiments and computational processing are necessary for the crucial task of structural and functional annotation of important genomic regions, such as genes or regulatory elements. Furthermore, many of the plant species whose genomes have been sequenced still lack sufficient experimental resources from these complementary analyses. These include the so called 'orphan crops', i.e. plants important for regional food and economic security that are not traded world-wide. Due to the lack of comprehensive transcriptomic, regulatory, or proteomic experiments we are limited in understanding their growth, senescence, yield, and responses to biotic and abiotic stresses[3–6], hampering the utilization of modern improvement tools, such as high-throughput (molecular) phenotyping, genomic selection, and genome editing. In light of this, novel approaches that can accurately predict gene annotations and regulatory genomic features directly from DNA sequences have the potential to provide valuable biological insights and assist in genomic editing applications. This is mainly because once an accurate predictive model for a particular biological process of interest has been developed, researchers can enjoy complete freedom to assess the effect of any variant. For instance, in the work of Rodriguez-Leal et al., the authors edited the tomato *CLAVATA3* gene (SlCLV3) promoter to increase fruit size and optimize inflorescence branching[7]. Due to the lack of functional annotations in the SlCLV3 promoter, they had to employ saturated promoter mutagenesis using the CRISPR/Cas9 system, followed by the selection of mutants with desirable fruit and inflorescence traits. An accurate predictive model could have potentially facilitated the identification of key regulatory elements in the SlCLV3 promoter enabling the implementation of a model-guided approach for promoter editing based purely on in silico-based analysis. Indeed, such approaches have demonstrated their feasibility and power in human genetics, successfully revealing variants, including rare alleles, underlying specific genetic diseases[8,9].

The complexity associated with sequence determinants of gene structure and regulatory features makes end-to-end deep learning-based approaches highly suitable for learning directly from DNA sequences to

[1]InstaDeep, London, UK. [2]Google DeepMind, London, UK. [3]These authors contributed equally: Javier Mendoza-Revilla, Evan Trop. [4]These authors jointly supervised this work: Thomas Pierrot, Marie Lopez. ✉e-mail: t.pierrot@instadeep.com; m.lopez@instadeep.com

achieve accurate predictions of specific outcomes. The majority of deep learning approaches have heavily relied on supervised learning to train neural networks in a task specific manner[10–14]. While successful across various applications, this approach depends on abundant labeled data, which can lead to sub-optimal performance and limited usability, especially in scenarios with data scarcity. Obtaining high-quality labeled data is often a time-consuming and costly process. To address this limitation, an approach called self-supervised learning, where a model is first trained on a large unlabeled corpus of data to learn discriminative features and subsequently fine-tuned on supervised tasks, has shown success across various fields, especially in natural language processing (NLP). Models such as BERT and GPT[15,16], specifically designed to handle sequential, discrete data within this self-supervised framework, have gained significant traction. Referred to as language models (LMs), they have also found widespread adoption in the field of biology[17–26]. These models offer the capability to be trained on unlabeled data and generate versatile representations capable of solving specific tasks.

LMs overcome another limitation of supervised learning, as they are not reliant on single reference genomes, which often provide an incomplete and biased genomic diversity depiction from a limited number of individuals. LMs can leverage multiple reference genomes, including those from genetically distant species, thereby increasing overall diversity, which has been shown to significantly enhance prediction performance[23]. This diversity is particularly relevant in plant species due to the structural complexities of their genomes, which hinder accurate mapping of polymorphisms across whole-genome alignments. Moreover, depending on the plant species under study, there may be a lack of large and representative collections of individual genomes, necessitating the use of genomic diversity from other species to improve prediction performance. Finally, LMs are well-suited for zero-shot learning, a transfer learning approach that enables the model to recognize and classify samples from new classes not encountered during training. This ability stems from their generalizability and comprehensive language understanding[27]. Zero-shot predictions represent an alternative approach to the traditional method of training supervised models on large amounts of functional genomic data. Since LMs are trained solely on DNA sequences, zero-shot predictions can be readily obtained, even for understudied plant species like orphan crops, in the absence of such functional genomic data. In the context of plant genomic research, only two LMs have been developed so far. Nonetheless, one of these was exclusively evaluated for its capability to predict the functional impact of genetic variants in *Arabidopsis thaliana*[28], while the other focused solely on predicting gene expression levels in various tissues of maize (*Zea mays*)[29].

In this study, we introduce a novel DNA large language model called the Agronomic Nucleotide Transformer AgroNT), which is based on the transformer architecture. AgroNT was trained using reference genomes from 48 plant species, with a primary focus on edible plant species. We assessed the performance of AgroNT across several prediction tasks ranging from regulatory features, RNA processing, and gene expression, and show that AgroNT can obtain state-of-the-art performance. We also illustrate the capability of AgroNT to improve prioritization of functional variants through zero-shot prediction. By exploiting the scalability of AgroNT, we further characterized the mutation space of promoter sequences and enhancer regulatory regions in the agriculturally significant 'orphan crop', *Manihot esculenta*, commonly known as cassava. Through in silico saturation mutagenesis we profiled over 10 million single point mutations. As a major contribution of this study, we provide the predicted impact of these variants for future research. Furthermore, we have made the pre-trained AgroNT model available for future research. Given the lack of comprehensive benchmarks in the context of deep learning-based methods in plant genomic research, we propose the use of the datasets covering several distinct genomic prediction tasks compiled here as the Plant Genomic Benchmark (PGB). Our aim is to enhance and provide a comprehensive assessment of the performance of deep learning methods in plant genomic studies. Overall, the results of our study emphasize the significant capabilities of AgroNT in the field of plant genomics and highlight its particular relevance for under-researched plant species.

## Results

### AgroNT: a novel large language model that integrates genomes across plants species

We developed a transformer-based DNA language model named the Agronomic Nucleotide Transformer (AgroNT), which learned general nucleotide sequence representations from genomic DNA sequences of 48 different plant species (Fig. 1a, Supplementary Fig. 1 and Supplementary Table 1; Methods). Building upon our previous work[23], our pre-training strategy involves performing masked language modeling (MLM) on a DNA sequence consisting of ~ 6000 base pairs (bp). Our tokenization algorithm splits the DNA sequence into 6-mers, treating each 6-mer as a token, and masks 15% of the tokens for prediction (Fig. 1b; Methods). For our fine-tuning strategy, we implemented parameter-efficient fine-tuning using the IA3 technique[30]. In this approach, we replaced the language model head with a prediction head, using either a classification or regression head based on the task. We kept the weights of the transformer layers and embedding layers frozen, or alternatively, unfroze a small number of the final layers to reduce training time for specific downstream tasks (Fig. 1c; Methods).

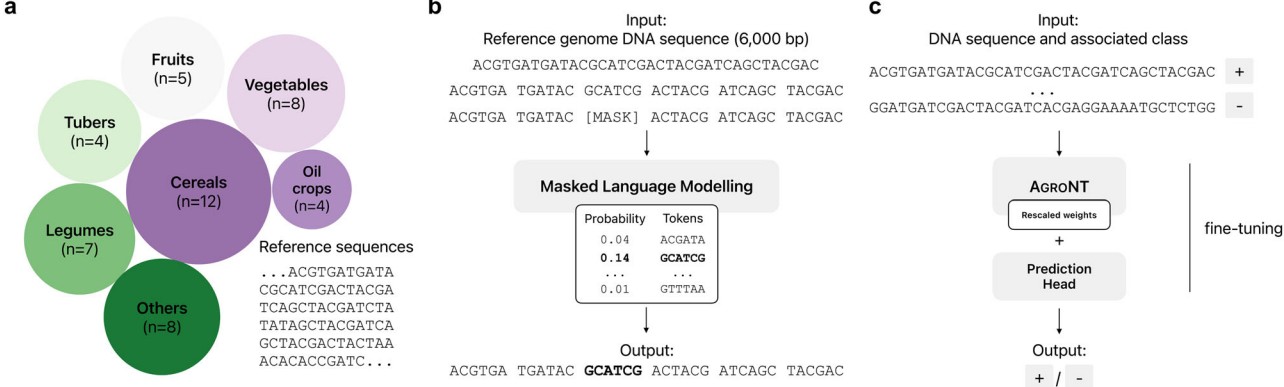

**Fig. 1 | Overview of agronomic nucleotide transformer. a** Data collection of plant species' reference genomes used for pre-training. The size of the circles is proportional to the number of species in each category. **b** Pre-training strategy. The input is a ~ 6000 base pair (bp) DNA sequence randomly selected from 48 plant species' reference genomes. Each sequence is tokenized into a 6-mer (i.e., a string of 6 nucleotides), and 15% of these tokens are randomly masked. The objective of masked language modeling (MLM) is to obtain the model parameters that best predict the masked tokens. **c** Fine-tuning strategy. After pre-training, we fine-tuned AgroNT using a supervised learning approach. We employed the parameter-efficient fine-tuning method, IA3, which introduces re-scaling weight vectors at the model's activations, enabling significant performance improvements while reducing computational resources compared to full model fine-tuning.

We introduced new learnable weights for the fine-tuning process. Throughout the study, we evaluated the performance of AgroNT across 8 genomic prediction tasks (Table 1) using exclusively our fine-tuning method and compared the obtained performance to other state-of-the-art methods, where available.

**Table 1 | Summary of the Plants Genomic Benchmark (PGB)**

| Task | No. datasets | No. classes/ Regression | Sequence length (bp) | Source |
|---|---|---|---|---|
| Polyadenylation | 6 | 2 | 400 | This study |
| Splicing site | 2 | 2 | 398 | Baten et al. |
| LncRNA | 6 | 2 | 101–6000 | This study |
| Promoter strength | 2 | Regression | 170 | Jores et al. |
| Terminator strength | 2 | Regression | 170 | Gorjifard et al. |
| Chromatin accessibility | 7 | 9-19 | 1000 | Zhao et al. |
| Gene expression | 6 | Regression | 6000 | This study |
| Enhancer region | 1 | 2 | 1000 | This study |

The table details the number of datasets, supervised learning task, sequence length, and source.

## Polyadenylation site, splice site, and long non-coding RNA prediction

We first evaluated the predictive capability of AgroNT in predicting alternative polyadenylation (APA), a process characterized by the selection of different polyadenylation sites within the same gene. APA has been recognized as a key regulator of gene expression in various eukaryotes, including plants[31]. For this analysis, we leveraged PlantAPAdb, a manually curated resource that offers an extensive catalog of APA sites in plants encompassing diverse biological samples[32]. The compiled dataset included more than 240,000 APA sites distributed across different genomic features from reference annotations in five plant species (Fig. 2a). As expected, the majority of polyadenylation sites are located at annotated 3′ UTR regions across plant species. AgroNT demonstrated high accuracy in predicting APA sites across various plant species. It achieved area under the receiver operating characteristic (ROC) curve (AUC) values ranging from 0.89 to 0.96 and area under the precision-recall curve (AUPRC) values ranging from 0.82 to 0.93 (Fig. 2b). To the best of our knowledge, no previous study has used PlantAPAdb to develop a method for predicting APA sites. Hence, for a comparative analysis of AgroNT's APA prediction performance, we trained the model on an APA dataset specifically composed of 3′UTR sequences from *Arabidopsis thaliana*, which was used to train the CNN-based model DeepPolyA[33]. Remarkably, AgroNT achieved an almost perfect performance with an AUC of 0.99, surpassing DeepPolyA's reported AUC of 0.97, as well as outperforming other methods such as DanQ and Deep-SEA, which obtained AUC values below 0.97 on the same dataset[33].

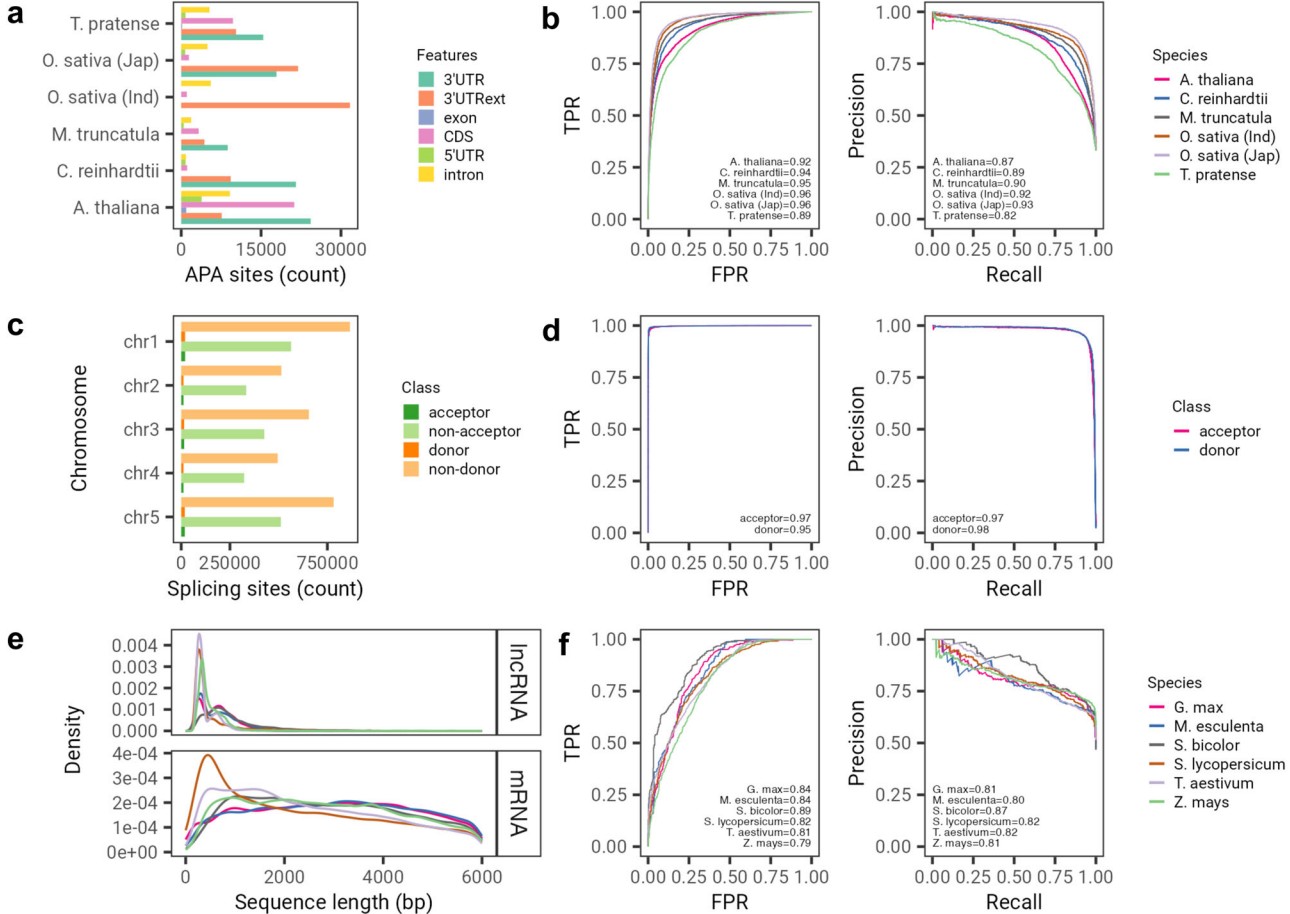

**Fig. 2 | AgroNT predicts polyadenylation, splicing, and long non-coding RNAs.**
**a** Distribution of alternative polyadenylation (APA) sites across different genomic features in 5 species as annotated in the PlantAPAdb database. 3′UTRext refers to the 3′UTR extended annotation. See[32] for further details on the annotation of APA sites. **b** Receiver operating characteristic curve (left) and precision-recall curve (right) for APA prediction. **c** Distribution of splicing sites (acceptor or donors) and non-splicing sites across the genome in *Arabidopsis thaliana*. **d** Same as in (**b**), but for splicing site prediction. **e** Distribution of the length of long non-coding RNAs and mRNA sequences across 6 species from the GreeNC database. **f** Same as in (**b**), but for long non-coding RNA prediction.

We hypothesize that the presence of exclusively introns, CDS, and 5′ UTR sequences in the negative training dataset of DeepPolyA, without any nearby or non-APA 3′ UTR signals, contributed to the reduced complexity of the classification task. This could explain the close-to-perfect performance obtained by AgroNT.

Next, we evaluated the performance of AgroNT in predicting splice sites, which are crucial contributors to transcriptome diversification in both animals and plants[34,35]. For this assessment, we used a dataset encompassing acceptor and donor sites, as well as non-acceptor and non-donor sequences across the entire genome of *Arabidopsis thaliana* (Fig. 2c). Notably, AgroNT demonstrated high performance in predicting both acceptor and donor sites, achieving AUROC values of 0.97 and 0.95, respectively. The AUPRC for acceptor and donor site prediction was 0.97 and 0.98, respectively (Fig. 2d).

We also evaluated the classification performance of AgroNT in distinguishing long non-coding RNAs (lncRNAs) from messenger RNAs (mRNAs). LncRNAs in plant species have been recognized for their significant roles in growth, development, responses to biotic and abiotic stresses, and the regulation of cell differentiation[36]. To assess the classification capability of AgroNT, we used a comprehensive repository of annotated lncRNAs in plants[37] and applied it to classify lncRNAs and annotated mRNAs across six plant species. The length of the two sequence types differed significantly, with lncRNAs having a median length of 457 bp, while mRNAs had a median sequence length of 2604 bp (Wilcoxon rank sum test, $P$ value $< 2.2 \times 10^{-16}$) (Fig. 2e). To mitigate the effect of the model relying solely on sequence length and encourage it to learn more meaningful features, we opted to retain only mRNA sequences that matched the length and GC content of lncRNAs, resulting in a more challenging and matched dataset (see Methods). AgroNT demonstrated high classification performance across the six plant species tested, achieving AUROC values ranging from 0.79 to 0.89 and AUPRC values ranging from 0.80 to 0.87 (Fig. 2f). Performing additional independent resamplings of the matched number of mRNAs used as negative sequences for the models showed low variability, with a median range of the difference in AUROC values across resamplings ranging from 1.6 to 3.9 points and AUPRC values from 2.5 to 4.6 across species, demonstrating the robustness of the AgroNT models against variations in negative sequence selection (Supplementary Fig. 2). We found only one study that used the GreeNC database to develop a LSTM-based model for predicting plant lncRNAs, namely lncRNA-LSTM[38]. Therefore, we trained AgroNT on the dataset compiled in that study, which differed from our dataset by including only mRNAs from *Zea mays* and not matching by length or sequence content. AgroNT exhibited almost perfect performance with an AUC of 0.991, which was comparable to the reported performance of lncRNA-LSTM (0.9934). We hypothesize that the much higher performance obtained in this classification setting by AgroNT is primarily due to the less challenging dataset, which was not matched by length nor GC content.

## Promoter and terminator activity prediction

We also assessed the predictive performance of AgroNT in determining the regulatory activity of core promoters and terminators, which play a crucial role in governing transcription initiation and termination[39–41]. To do this, we leveraged two recently developed datasets that extensively examined the core promoters and terminators of three plant species. These promoters and terminators were analyzed through STARR-seq assays, which were individually conducted in transiently transformed tobacco leaves or maize protoplasts[40,41] (see Methods). We note that in the case of terminators, randomized sequences with varying GC content were also added in the study to evaluate the effect of GC content on terminator strength. Across the species, we observed higher activity in the tobacco system compared to the maize protoplasts for promoter strength, and lower activity in the tobacco system compared to the maize protoplasts for terminator strength, consistent with the observations in both studies[40,41] (Fig. 3a, b). In the case of terminators, the sequences with varying GC content also showed a much more restricted distribution compared to naturally occurring terminators.

AgroNT demonstrated the ability to predict the strength of promoters in held-out sequences with high performance, achieving an $R^2$ value of 0.7 and 0.73 in the maize system and tobacco model, respectively (Fig. 3c). Notably, AgroNT outperformed a CNN-based model aimed to predict promoter strength[40] across all plant species and systems, on average, by 3 $R^2$ points (Fig. 3e). In the case of terminators, AgroNT also demonstrated high performance in predicting their strength, achieving an $R^2$ value of 0.67 and 0.77 in the maize and tobacco systems, respectively (Fig. 3d). When compared to a second CNN-based model trained for predicting terminator strength[41], AgroNT either surpassed or matched the performance of the CNN-based model in 5 out of 6 datasets, and only slightly underperformed in one (Fig. 3f). We note that, it is possible that removing GC-based terminators from training could result in higher performance by allowing the model to better learn different types of motifs associated with terminator strength. Overall, these results demonstrate the capability of AgroNT to learn these two important regulatory elements and provide a strong model for developing optimized synthetic promoters and terminators in plant species.

## Chromatin accessibility prediction

As an additional example of regulatory sequence prediction, we also evaluated the performance of AgroNT in predicting genome-wide chromatin profiles. By analyzing chromatin profiles, it is possible to identify key regulatory elements, such as enhancers and promoters, where many putative causal variants in humans and plants are often found[42–44]. We leveraged a recent collection of ATAC-seq data from multiple tissues of six model and important crop species[11]. The dataset included an average of 81,000 open chromatin profiles across all the tissues for the six plant species, with the profiles distributed among various tissues showcasing the sparsity of these annotations across the genome. (Fig. 4a). AgroNT demonstrated strong performance in predicting chromatin profile peaks across different plant species and tissues, achieving AUROC values ranging from 0.94 to 0.98 (Fig. 4b) and AUPRC values ranging from 0.51 to 0.67 (Supplementary Fig. 3). We note that these performance metrics highlight the predictive capability of AgroNT in identifying these regulatory sequences, considering the low proportion of positive sequences in the datasets across plant species (average 0.12). Further, AgroNT either outperformed or achieved comparable performance to a CNN-based based model, plantDeepSEA[11], in five out of the six tested plant species when comparing the performance across all tissues within each plant. On average AgroNT exhibited a marginally higher performance, i.e. less than 1 AUROC or AUPRC point variation. It is worth noting that these predictions were made using a limited sequence length of 1000 bp for predicting chromatin profiles, ensuring a fairer comparison against the CNN-based model. We anticipate that by extending the sequence context length to 6000 bp length, the capability of AgroNT, the model's performance is likely to further improve[45,46].

We also evaluated whether these models could be used to aid in detecting causal variants by evaluating a gene with a well-established role in crop yield, as recently done in ref. 11. Specifically, we evaluated the effect of nine non-coding variants of the *DEP1* gene that have been shown to affect leaf morphology and yield in *O. sativa*[47,48]. We found that a single variant, vg0916410299 (C/G), had the strongest effect on chromatin accessibility across multiple tissues, with the strongest effect observed in roots and leaves (Supplementary Fig. 4). Interestingly, an in silico saturation mutagenesis analysis showed that while vg0916410299 had the strongest effect compared to all nearby variants, the effect of the single reported alternative allele G in RiceVarMap2[49] was slightly lower than that of the A allele, a non-segregating allele (Supplementary Fig. 5). This result illustrates how the developed models can not only help pinpoint putative non-coding causal variants, but also how they can be used to highlight other potential mutations, whether segregating or not, that may be relevant for gene editing purposes.

## Tissue-specific gene expression prediction

To assess the performance of AgroNT in predicting the levels of gene expression across tissues, we collected and processed data from gene

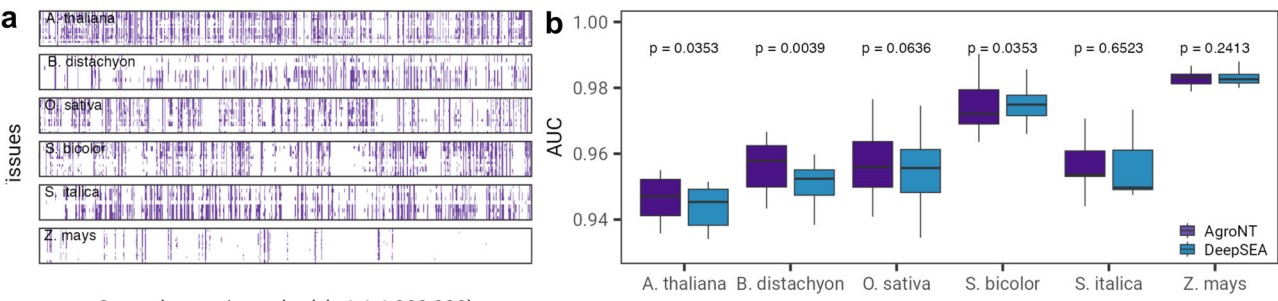

**Fig. 3 | AgroNT predicts promoter and terminator activity. a** Promoter and terminator (**b**) strength distribution derived from STARR-seq assays conducted in transiently transformed maize protoplasts and tobacco leaves. **c** AgroNT prediction of promoter and terminator (**d**) strength. **e** The performance of AgroNT in predicting promoter and terminators (**f**) compared to two different CNN-based models across plant species. In the case of terminators, randomized sequences with varying GC content were also included in the analyses.

**Fig. 4 | AgroNT predicts genome accessibility across species and tissues. a** Open chromatin profiles across tissues. The heatmap illustrates the genomic position of chromatin profiles from six plant species across tissues in a 1-megabase window of chromosome 1. Each cell in the heatmap represents a 1000-base pair (bp) region. The training strategy of AgroNT involved detecting chromatin profiles in the center of a 1000 bp region. **b** The performance of AgroNT in predicting chromatin profiles across species and tissues is measured using the area under the curve (AUC) of the receiver operating characteristic curve. *P* values shown are based on a two-sided Wilcoxon signed test comparing the performance of AgroNT with a model based on the DeepSEA architecture.

expression atlases of five plant species. Our compiled dataset included tissue-specific gene expression values that ranged from 7 tissues in rice and up to 56 tissues in *Arabidopsis thalaiana* (Methods). To predict gene expression, we trained AgroNT on promoter-proximal sequences based on transcription starting sites (TSS) annotations and ensured that the sequence overlapped slightly with the coding sequence, as this has been shown to aid in gene expression prediction accuracy across different species, including plants[50]. Specifically, for each gene we extracted 5000 bp upstream and 1000 bp downstream of the TSS, and used these sequences as input for our models. To prevent potential data leakage, the training splits were based on

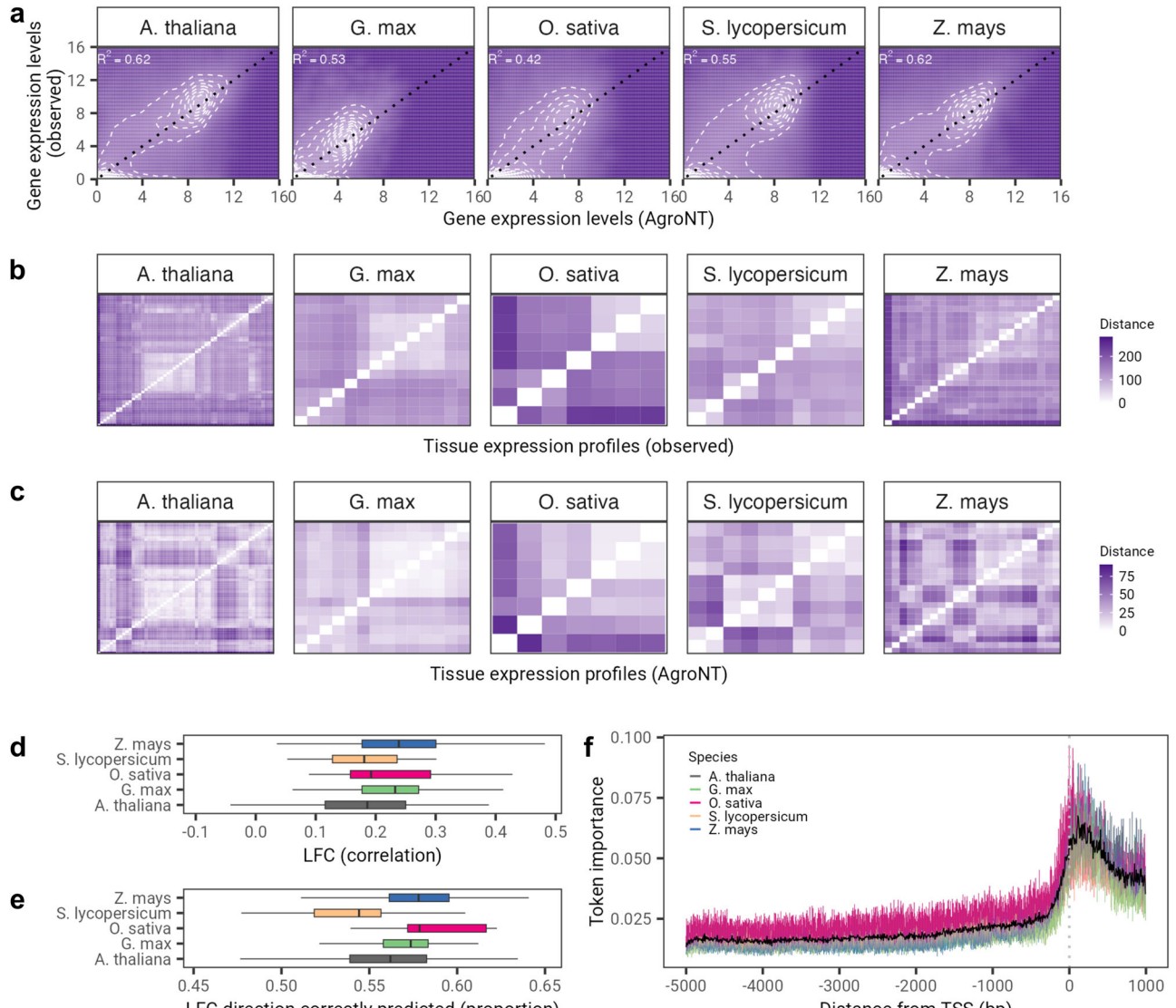

**Fig. 5 | AgroNT provides gene expression prediction across different plant species. a** Gene expression prediction on holdout genes across all tissues are correlated with observed gene expression levels. The coefficient of determination ($R^2$) from a linear model between predicted and observed values are shown. **b**, **c** Tissue expression profiles between observed (upper panels) and predicted gene expression levels (lower panels) on holdout genes. The order of the cells in the heatmap is based on a hierarchical clustering using the euclidean distance of the observed gene expression levels across tissues. **d** Correlation between observed and predicted log fold changes (LFC) for all pairwise tissue comparisons (upper plot) and (**e**) LFC directions, i.e. accuracy of the model in predicting upregulated or downregulated genes between tissues. **f** Token importance analysis across promoter-proximal sequences. Token importance is computed as the absolute difference between the original model prediction and the predicted gene expression level after inserting a random token across the promoter-proximal sequence. Token importance is averaged across tissue for each species. Black line denotes the mean value across species. Vertical dotted line indicates the transcription starting site (TSS) position.

assigning promoter-proximal sequences to gene families, and gene families were randomly allocated to training splits, as recently conducted in ref. [51]. AgroNT obtains moderate prediction performance of gene expression levels for genes across the five plants species and across all tissues ($R^2 = 0.419 - 0.621$) (Fig. 5a). The prediction performance was relatively consistent across individual tissues with the exception of pollen which had consistently lower prediction performance across plant species (Supplementary Tables 2–6). This might be related to the inaccessible nature of pollen developing in the anther and the resistant pollen wall that might affect proper pollen isolation, as recently suggested[52]. Considering the absence of models capable of predicting quantitative gene expression levels, and to conduct a comparative analysis for this task, we also trained AgroNT to predict whether genes are expressed or not. To accomplish this, we employed a dataset derived from a collection of *Zea mays* tissues, which was previously utilized by a recent study to train a CNN-based model[51]. Notably, AgroNT achieved superior performance compared to the model trained on

promoter sequences (AUROC 0.89 versus 0.81) as well as terminator sequences (0.89 versus 0.82).

To further explore the capabilities of AgroNT for gene expression prediction, we also evaluated the performance of AgroNT in reproducing tissue-specific expression patterns. To do this, we conducted hierarchical clustering using the distances between observed gene expression levels across tissues. We then assessed how well the predicted distances replicated these patterns (Fig. 5b, c and Supplementary Fig. 6–10). Encouragingly, AgroNT successfully reproduced the general patterns of similarity across tissues. Nonetheless, we observed that the predicted distances were significantly shorter than the observed distances, indicating that, on average, the predicted gene expression levels were more similar across tissues than the observed. To further evaluate the tissue-specific performance, we also computed log-fold changes (LFC) for all pairs of tissues and assessed the correlation between the observed and predicted LFC values for each species separately (Fig. 5d). The correlation between the observed and predicted

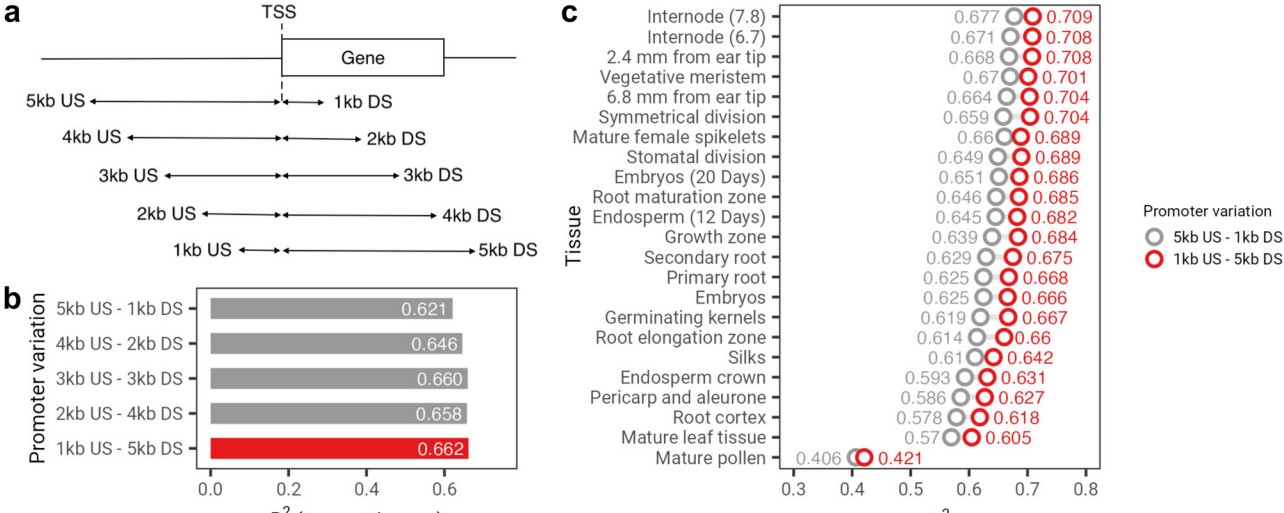

**Fig. 6 | Assessing various types of promoter sequences as input for gene expression prediction in Z. mays. a** Schematic representation of the strategy employed to extract various types of promoter sequences based on different upstream and downstream sequences relative to the transcription starting site (TSS). **b** Performance of gene expression prediction in *Z. mays* across all 23 tissues for five different types of promoter sequences. **c** Performance of gene expression prediction of the model based on promoter sequences derived from 5000 bp upstream and 1000 bp downstream versus 1000 bp upstream and 5000 bp downstream of the TSS. Acronyms: US stands for upstream. DS stands for downstream.

LFC values was relatively low, with similar correlation values observed across species (ranging from 0.18 to 0.24). Similarly, when examining the predicted LFC directions, which indicate the accuracy of the model in predicting whether genes are upregulated or downregulated between tissues, we found that slightly over half of these predictions were correct (with a median ranging from 0.54 to 0.58) (Fig. 5e). It is likely that the model has primarily learned to recognize essential features associated with basal transcriptional activity. Indeed, when evaluating the importance of specific regions within the prediction sequences, we found that sequences positioned approximately 100 base pairs from the annotated TSS had the most significant influence on the predictions (Fig. 5f). This observation is consistent with the understanding that these sequences serve primarily as binding sites for transcription factors that are crucial for basal transcription levels, independent of tissue or cell types. This result also illustrates the challenge of accurately predicting tissue specificity based on promoter-proximal sequences only. The model's inability to capture tissue-specific regulatory regions, such as enhancers that can be located far away from the promoter region and exhibit more variable and cell type-specific activity, contributes to this difficulty.

Lastly, and motivated by the higher importance attributed by our model to regions further downstream than the TSS (Fig. 5f), we generated additional promoter-proximal sequences by sliding 1000 bp increments: 4000 bp upstream and 2000 bp downstream, 3000 bp upstream and downstream, 2000 bp upstream and 4000 bp downstream, and 1000 bp upstream and 5000 bp downstream of the TSS (Fig. 6a). We limited this additional analysis to *Zea mays*, as our model achieved the highest performance in this species among the five plant species tested and had the second-largest tissue collection (23 tissues). Notably, the model trained on sequences extending 5000 bp downstream of the TSS obtained an $R^2 = 0.662$, representing a 4% increase compared to our previous model trained on sequences including only 1000 bp downstream of the TSS (Fig. 6b). This improvement was consistent across all tissues (Fig. 6c) and aligns with recent observations highlighting the importance of the entire gene regulatory structure in determining gene expression levels[50]. Furthermore, given the median length of *Zea mays* genes in the dataset used is 2410 bp, it is likely that the model trained on sequences extending 5000 bp downstream of the TSS has also learned to leverage other regulatory regions, such as terminators, for improved gene expression prediction. These results underscore the importance of approaches for model interpretation and the

significance of evaluating the most suitable input sequences to develop more powerful models.

## Zero-shot prediction of functional variants

AgroNT can also be leveraged for zero-shot learning, a transfer learning approach that enables the model to recognize and classify samples from new classes that were not encountered during training. Hence, we evaluated the ability of AgroNT to assess the functional consequences of different types of genetic variants through various zero-shot scores. Specifically, we computed zero-shot scores by considering different aspects of vector distances in the embedding space, along with a score derived from the loss function. Specifically, we calculated the log-likelihood ratio (LLR) between the probabilities of alternate and reference alleles in a given sequence (see Methods). As a baseline for comparison, we also included a LLR score called the Genomic Pre-trained Network (GPN) score[28]. Like our approach, this score is based on a multi-species model pre-trained on reference genomes of *Arabidopsis thaliana* and seven related species within the Brassicales order. In addition to these, we also included two conservation scores, phastCons[53] and phyloP[54], in the comparison. These scores assess the evolutionary constraint on specific genomic positions and provide valuable information regarding potential functional implications.

We first investigated the utility of zero-shot scores in evaluating deleteriousness. The presence of aggregated genomes from the 1001 Arabidopsis Genome Project[55] and the recently compiled RiceVarMap2 database[49], provided an opportunity to assess the impact of natural selection on missense and synonymous mutations across a range of allele frequencies for *Arabidopsis thaliana* and *Oryza sativa*, respectively. When comparing the ratio of missense to synonymous variants, we observed that, in both species, very rare variants (frequency lower than 0.1%) and rare variants (frequency between 0.1% and 1%) exhibited the highest missense/synonymous ratio compared to the expected ratio of 1 for non-deleterious variants (Fig. 7a). As allele frequency increased, the missense to synonymous ratio decreased consistently, suggesting that a considerable proportion of missense variants with frequencies lower than 1% may have the strongest deleterious effects in both species. Based on this observation, we examined the enrichment of missense variants in these two low frequency categories compared to common variants (frequency > 5%) using odds ratios (OR) across different quantiles of the score distributions (Fig. 7b). We note that in the case of *Oryza sativa*, the missense to synonymous ratio remained above 1 even for

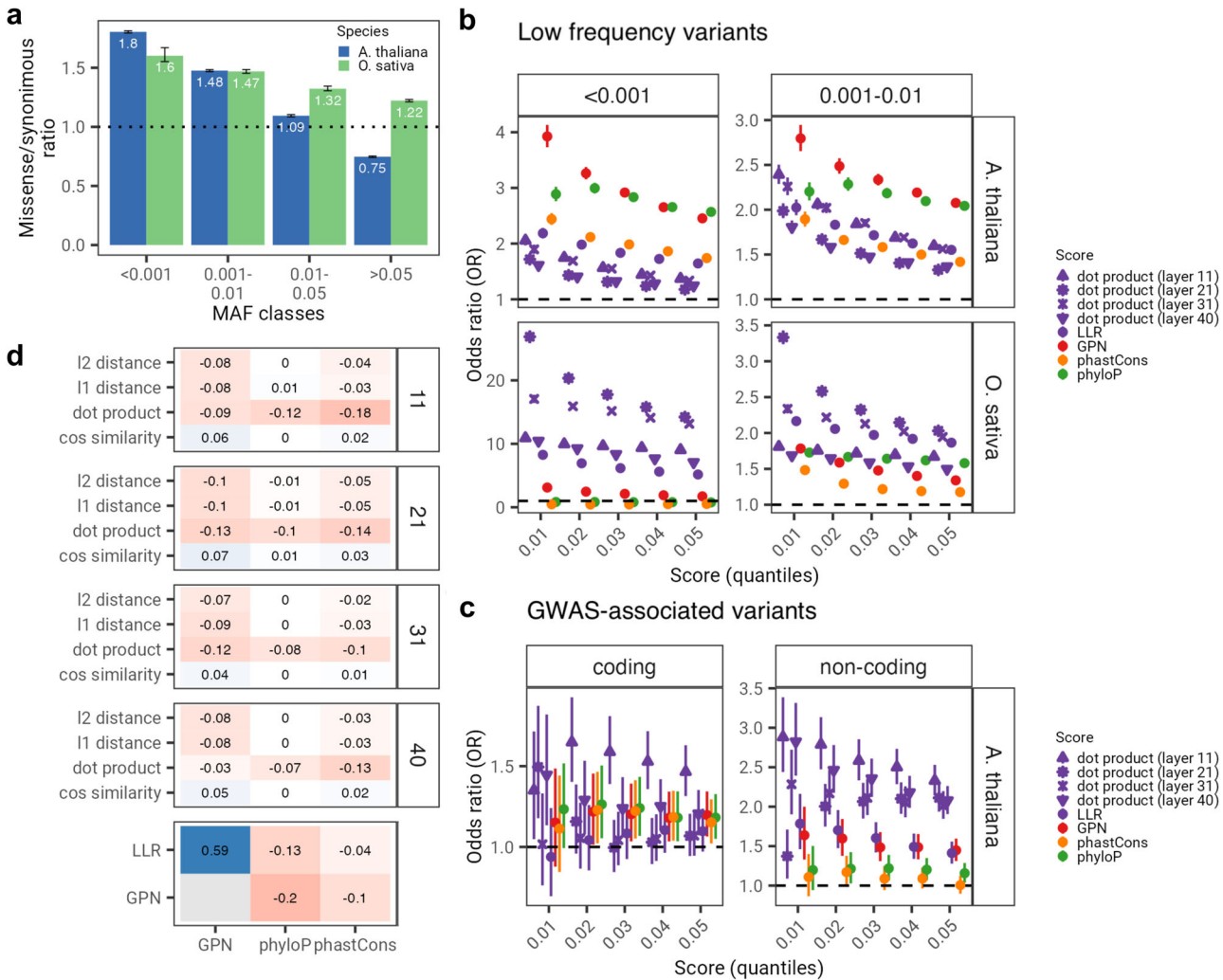

**Fig. 7 | Zero-shot based prediction for variants of different functional classes.**
**a** Ratio of missense (i.e. non-synonymous) to synonymous variants in the 10001
Arabidopsis thaliana Genome Project and RiceVarMap2 dataset across frequency
classes. Error bars denote 95% confidence intervals based on 100 chromosome-
based samplings with replacement. **b** Odd ratios (OR) across different score
thresholds used to assess enrichment of very low frequency (i.e. potentially dele-
terious) variants. **c** OR across different score thresholds used to assess enrichment

coding and non-coding GWAS-associated variants. Error bars denote 95% con-
fidence intervals. Dashed black line indicates an OR of 1. **d** Correlations between
AgroNT zero-shot based scores and conservation (phyloP and phastCons) and GPN
scores. Facets indicate the layer used to compute the zero-shot scores. LLR refers to
the log-likelihood ratio between the alternate and reference allele probabilities from
AgroNT.

common variants, suggesting a heavier genetic load in this species, and a
higher proportion of potentially deleterious variants even at common fre-
quencies. Nonetheless, given that this analysis is based on a completely zero-
shot manner, and no supervised model is being trained considering com-
mon variants as non-deleterious, the result arising from this comparison is
still informative regarding the enrichment of deleterious variants on the tail
of the distribution scores.

In our analysis examining the informativeness of zero-shot scores for
discerning potential deleterious variants, we found that, in *Arabidopsis
thaliana*, GPN and phyloP showed the highest ORs for both frequency
classes and across quantiles (Fig. 7b). The LLR score derived from AgroNT
also exhibited significant ORs, particularly at the lowest quantiles, and
slightly surpassed those of phastCons. The dot product of embeddings
scores (across four layers) from AgroNT also exhibited significant ORs,
surpasing the ORs from phastCons. Other zero-shot scores based on the
embedding space and other layers did not exhibit superior performance
either, except for the cosine similarity score, which is correlated with the dot
product score (Supplementary Fig. 11). Considering that GPN is based on
pre-trained sequences from Brassicales-related species, it is likely that the
model leverages this variation to better capture the frequency of variants in

*Arabidopsis thaliana*, thereby aiding in the detection of these rare and likely
deleterious variants. Indeed, across all variants, GPN scores displayed a
stronger correlation with minor allele frequency in *Arabidopsis thaliana*
($r = 0.07$, $P$ value $< 2.2 \times 10^{-16}$) compared to LLR from AgroNT ($r = 0.05$, $P$
value $< 2.2 \times 10^{-16}$), which was trained on multiple plant species. In line with
this observation, we observed that GPN exhibited significantly lower ORs,
for both frequency classes and across quantiles, than scores derived from
AgroNT for *Oryza sativa* variants. For the very rare variants ($< 0.1\%$), in
particular, the scores obtained based on the dot product of the embeddings,
as well the LLR from AgroNT, were up to 8-times higher than the GPN
scores, a difference that was not observed when comparing the GPN score
versus AgroNT scores in *Arabidopsis thaliana*. This illustrates the advantage
of pre-trained models that are based on genomes from diverse set of species,
compared to those trained on more closely-related species. While GPN was
not trained on any species related to *Oryza sativa*, it is notable that GPN still
exhibited significant ORs for both frequency classes and across quantiles,
surpassing those of the conservation scores. This highlights a degree of
transferability of GPN to other plant species.

We next evaluated the performance of these scores to identify genome-
wide associated variants (GWAS) based on the AraGWAS Catalog[56], a

manually curated database for standardised GWAS results for *Arabidopsis thaliana*. Considering that the variants analysed here are determined exclusively from annotations obtained from *Arabidopsis thaliana*, we regard this comparison as challenging when compared to GPN. We examined the enrichment of coding and non-coding GWAS variants relative to non-GWAS variants (Fig. 7c). Across quantiles, LLR achieved a similar performance to GPN for non-coding variants, and lower performance for coding variants. It is possible that, given that AgroNT was pre-trained across multiple plants, and that the number of non-coding sequences observed by the model compared to coding sequences was therefore higher, AgroNT constructs better representations for these class of variants. In line with the results from low frequency variants, we also observed that scores derived from the embedding space, such as the dot product, exhibited high ORs for both coding and non-coding GWAS variants, demonstrating its utility to capture the effect of these variants (Fig. 7c and Supplementary Fig. 12). Finally, we also observed that the zero-shot scores derived from AgroNT exhibited a very low correlation with the conservation scores, indicating that they capture genomic information beyond just the degree of conservation (Fig. 7d). This observation suggests the possibility of integrating these scores to create a more robust annotator capable of predicting the potential impact of genetic variations in plant genomes.

### Profiling the regulatory potential via in silico mutagenesis

Lastly, we leveraged the capability of AgroNT to obtain accurate predictions across a variety of regulatory features by further training AgroNT to predict intergenic enhancer elements and gene expression levels across 11 tissue/organ types in the *Manihot esculenta* (cassava) genome, and performed in silico saturation mutagensisis analyses (see Methods). AgroNT demonstrated strong performance in detecting intergenic enhancer sequences, achieving an AUC value of 0.88 (Fig. 8a). Through a large-scale in silico mutagenesis analysis, we systematically mutated each site to all three possible distinct nucleotides across the central 400 bp region of the top 1000 most confidently predicted enhancer regions by AgroNT (probabilities ranging 0.975 to 0.999). As a result, mutating the nucleotides within these regions is more likely to produce an effect that AgroNT can accurately identify. By estimating the log-fold change (LFC) between the probability of the mutated sequence and that of the original sequence, we note, as expected, that the majority of mutations do not strongly influence the predicted probability of an enhancer sequence (median absolute LFC probability $= -1.10 \times 10^{-3}$) (Fig. 8b), since the majority of the evaluated mutations are likely not impacting relevant enhancer functionality. To further assess the impact of the evaluated mutations we identified potentially strongly-affecting mutations as those with a LFC probability less than $-0.042$ (0.01% quantile), implying a disruption of enhancer elements. This resulted in 120 mutations which impacted 15 distinct enhancer sequences. Notably, the most potent mutation overlapped the TCGAAT and DOF motifs (Fig. 8c), while the second most influential mutation targeted the DOF motif (Fig. 8d). This observation suggests that the AgroNT-trained model has likely acquired an understanding of the significance of these motifs in relation to enhacer regions. To assess the impact of mutations on promoter-proximal regions, we also trained AgroNT to predict gene expression levels across 11 tissue/organ types in cassava, achieving moderate predictive performance ($R^2 = 0.51$, $P$ value $< 2.2 \times 10^{-16}$ across all tissues) (Fig. 8e and Supplementary Fig. 13). Similar to the enhancer-based mutagenesis analysis, we then calculated the LFC between the predicted gene expression levels of the mutated sequence and the original sequence for the 1000 promoter sequences derived from genes with the highest average gene expression levels across all tissues (Fig. 8f). Focusing on mutations with the most substantial impact, specifically those with LFC less than $-0.28$ (0.01% quantile), we identified 136 mutations across 20 distinct promoter sequences. The most significant mutation affected the Trihelix motif (Fig. 8g), while the second most significant mutation impacted the DOF motif (Fig. 8h). These findings suggest that the AgroNT-trained model has also developed an understanding of the importance of these motifs in relation to gene expression levels. We have provided the predicted impact of

all possible single-nucleotide substitutions across all enhancer and promoter sequences for future research purposes.

## Discussion

In this paper, we showcase the capabilities of AgroNT, a novel DNA-based LLM, for accurately making predictions on a wide range of genomic tasks across various plant species. Due to its pre-training on a large collection of plant genomes, AgroNT inherently possesses transfer learning capabilities, allowing it to aggregate knowledge from all assembled plant genomes, and enhancing its predictive power. In the context of regulatory-based prediction tasks, it is reasonable to hypothesize that AgroNT has learned to capture genomic regulatory features that likely hold functional importance across plant species, leading to improved performance. Recent studies have indicated that many key promoter elements identified in animals are also enriched in plant promoters[57–60], suggesting that leveraging more phylogenetically-distinct species could potentially offer further performance improvements. AgroNT differs from the majority of previously developed models that rely on training with specific genomic sequences from individual plant species and use distinct deep learning architectures for each prediction task. In contrast, we exclusively employed a parameter-efficient fine-tuning approach for all predictive tasks. This flexible and effective strategy either matched or outperformed current state-of-the-art methods in the assessed tasks. It simplifies model selection, ensuring reliable high performance without the necessity for extensive architecture analysis for each prediction task.

Several paths to further improve model performance appear promising, particularly in two areas with a direct impact on breeding engineering. The first of these relates to the prediction of functional and deleterious genetic variants. Several studies on domesticated plant species have found that domestication is often associated with an increase in harmful deleterious variants[61–64]. With low rates of sexual reproduction, these deleterious variants are maintained in the population and can hinder breeding efforts[65]. As such, confidently identifying these mutations could facilitate targeted decisions to purge genetic load from crop species and advance genetic gains[66]. Our analysis shows that AgroNT could be used to prioritize functional variants based solely on DNA sequences, which may be particularly relevant for understudied plant species, such as orphan crops, that currently lack functional genomic annotations. Interestingly, our analysis comparing zero-shot scores from AgroNT and the GPN model[28] in prioritizing deleterious suggests that leveraging the genetic variability from species of relatively short phylogenetic distance during the pre-training leads to better performance in detecting these variants. As a direct application, the recent study of Long et al.[67], which assembled 52 *Euphorbiaceae* genomes to compute evolutionary conservation scores and assess its use in detecting deleterious variants in cassava, could be compared against zero-shot scores derived from AgroNT after additional pre-training with these genomes. In that study, over half of all base pairs across orthologous genes had an alignment depth of > 31 species, meaning that only around half of the bases could be confidently assessed for their deleteriousness when computing alignment-based conservation scores. In contrast, AgroNT can leverage multiple genomes without having to rely on whether they have been aligned or not, and as such, can compute, for example, a zero-shot score for any variant. In addition to zero-shot scores, we envision using supervised models as another potential approach to enhance the detection of functional and deleterious variants. Current studies predominantly relying on sequence conservation metrics to predict deleterious mutations in plant species have shown varying performance (AUC ~ 0.5 to AUC > 0.9)[68,69]. Nonetheless, these methods still trail behind those developed for humans, suggesting an opportunity for improvement. Given the low correlation that we observe between the zero-shot based scores from AgroNT and conservation metrics, it is feasible to assume that their addition as features into the currently developed models may lead to an improvement in performance. Lastly, another potential avenue for improvement involves harnessing AgroNT's

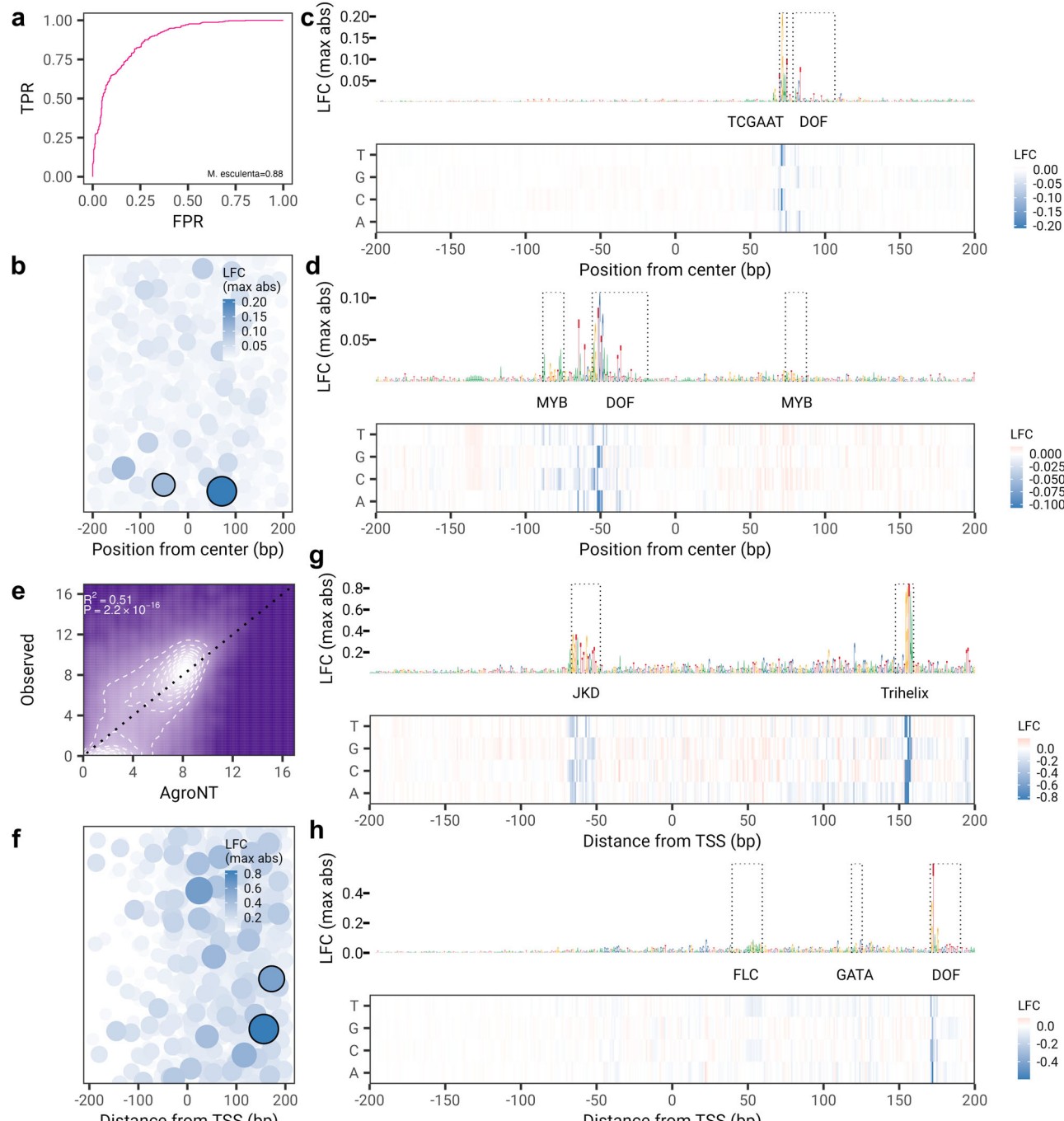

**Fig. 8 | In silico mutagenesis in the Cassava genome. a** Receiver operating characteristic curve for the prediction of intergenic enhancer regions. **b** The mutations with the strongest log-fold change (LFC) probabilities for each of the top 1000 enhancer regions are shown in relation to their distance from the central 400 bp region. The size of the points is proportional to the LFC value. Circled points represent the mutations with the strongest LFC values shown in (**c**) and (**d**). **c** The in silico mutagenesis map for the enhancer region with coordinates chr1:6334152-6335151. The bottom panel displays the LFC value for mutating each site to all three possible distinct nucleotides across the central 400 bp region. The upper panel features the DNA sequence logo, with the nucleotide heights representing the maximum absolute LFC value. Motif positions are indicated within a dotted rectangle, with the motif names displayed below. **d** Same as in (**c**), but for the enhancer region with coordinates chr2:15863252-15864251. **e** Gene expression prediction on holdout genes across all tissues. The coefficient of determination ($R^2$) from a linear model and associated *P* values between predicted and observed values are shown. **f** Same as in (**b**), but for LFC gene expression predictions across promoter sequences. **g** Same as in (**c**), but for promoter sequence with coordinates chr12:27521008-27527007. **h** Same as in (**c**), but for promoter sequence with coordinates chr15:862119-868118.

transfer learning capabilities when dealing with a lack of putative deleterious variants in a specific plant species. For instance, the recently introduced model PrimateAI-3D[70] has demonstrated that systematically cataloging common variants in non-human primates can accurately infer the pathogenicity of missense variants in humans. Similarly, utilizing

plant catalogs like the 1001 Arabidopsis Genome Project, one could develop a classifier to predict rare (or absent) variants, assuming their high enrichment in deleterious variants.

The second area with direct impact on breeding engineering relates to regulatory genomics. Cis-regulatory elements do not work in isolation. In

plant systems, it has been demonstrated that interactions between promoters and terminators affect gene expression, and the relative strength of a terminator depends on which promoter it is paired with[71]. While our gene expression prediction focused on promoter-proximal sequences, incorporating terminators could improve results, as observed in binary expression prediction in *Zea mays*[51]. An additional approach could also leverage distinct omics assays. Leveraging AgroNT, one could combine candidate regulatory elements based on chromatin accessibility assays with promoter sequences to enhance gene expression prediction. As demonstrated in our evaluated gene expression models, accurately predicting tissue-specific gene expression remains a challenge. Data based on gene regulation at single-cell resolution have the potential to unravel complex regulatory interactions and identify targets for genetic manipulation. In this vein, single-cell-based assays have recently been employed to study cell-type-dependent gene regulation in plants[72–75]. These studies identified cell-type-specific changes in gene expression and chromatin accessibility, illustrating the potential use of single-cell assays in prediction models. Encouragingly, at least based on single-cell assays from human cells, LLMs trained on single-cell data have shown high performance across numerous downstream tasks[76,77]. Nonetheless, given the current data sparsity of single-cell experiments and the difficulty in capturing sufficient cells of rarer types, at least in plants, leveraging this type of data in deep learning approaches will likely depend on novel methods that can significantly reduce the cost per cell. In addition to leveraging distinct assays, another avenue relates to expanding the sequence context to capture additional regulatory elements such as enhancers, silencers, and insulators[78,79]. Models like Enformer[45] and the recent HyenaDNA[80], which can handle larger regions and longer context lengths, represents yet another promising approach to better predict tissue-specific gene expression. Despite the potential approaches for improvement, as of now, AgroNT stands as the most accurate plant gene expression predictor. Comparative studies achieved $R^2$ values of 0.27[50] and 0.09-0.19[29] using promoter-proximal sequences to predict gene expression levels in *Arabidopsis thaliana* and *Zea mays*, respectively. In comparison to these two models and based on our accuracy on both *Arabidopsis thaliana* and *Zea mays*, AgroNT demonstrates up to a three-fold improvement in performance.

To aid the field of plant genomics, we have made the pre-trained AgroNT model available for future research, as well as the datasets compiled here, which we term the Plants Genomic Benchmark (PGB). Furthermore, we have released the effect predictions for promoter sequences and enhancer regulatory regions in the still understudied crop cassava (*Manihot esculenta*). Altogether, continued progress in deep learning-based models in plant genomics will open exciting opportunities to explore the expanding catalogs of genetic variants linked to critical plant traits, ultimately enabling the successful engineering of high-yielding and stress-resistant crops, ensuring food security and agricultural sustainability in the face of changing climatic conditions.

## Methods
### Model
In our study, we utilize Language Models (LMs) as the primary method for learning the foundational representations of plant genomes. LMs have demonstrated remarkable performance and generalization in various domains such as NLP, proteomics, and recently in our work in genomics[15,23,81]. Our previous work showed the ability of our pre-trained DNA LM's to match or out perform task specific models on variety of human genomics tasks. LM's, as statistical systems, serve to estimate the likelihood of a sequence of tokens occurring given some token vocabulary, where tokens could represent words, amino acids, or nucleotide k-mers. There are two main approaches to language modeling: masked language modeling (MLM), where a portion of the input sequence is masked and/or randomly replaced with another token[15], and causal language modeling (CLM), where a sequence of tokens is used to predict the next token[82]. More precisely, we use a deep learning model called transformers whose primary component, the self-attention mechanism, is capable of modeling global

dependencies within sequences[83]. Our training technique closely follows BERT, the Bidirectional Encoder Representation from Transformer, which employs MLM, thus allowing it to benefit from the self-attention bidirectional property[15].

### Architecture
Our model architecture can be described as an encoder only transformer that is comprised of one billion trainable parameters. The tokenizer's vocabulary consists of all possible 6-mers using the four nucleotides A, T, C and G, additional special tokens, and N to represent all unknown nucleotides in a DNA sequence. The tokenization algorithm works by first splitting genomic DNA sequences on N's and subsequently identifying 6-mers in the chunked sequences. The tokenization strategy relies exclusively on non-overlapping 6-mers. The model takes as input a sequence of 1025 tokens (including the class token) which are passed into an embedding layer that projects each token into a 1500 dimensional space and are subsequently added to a learned positional embedding. This combined embedding vector is then fed through 40 attention blocks, with each attention block consisting of a layer normalization layer, multi-headed attention layer, another layer normalizatino layer, and multi-layer perceptron (MLP). The final layer, a Roberta LM head[84], transforms the final attention layer output into a distribution over the tokenizer's vocabulary at each position of the input sequence. See Supplementary Table 1 for model architecture details and Supplementary Fig. 1 for model schematic.

### Pre-training dataset
Our pre-training dataset was built from (mostly) edible plant reference genomes contained in the Ensembl Plants database[85]. The dataset consists of approximately 10.5 million genomic sequences across 48 different species. See Supplementary Data 1 for the dataset details, including genome annotation version and source. For data preparation, we followed the strategy employed in our previous work[23]. Once we obtained the reference genome FASTA for each species, we assembled them into a single FASTA file. In this FASTA file, all nucleotides other than A, T, C, G were replaced by N. We used a tokenizer to convert strings of letters into sequences of tokens. The tokenizer's alphabet consisted of the $4^6 = 4096$ possible 6-mer combinations obtained by combining A, T, C, G, as well as five additional tokens representing standalone A, T, C, G, and N. It also included four special tokens: the padding [PAD], masking [MASK], the beginning of sequence (also called class; [CLS]), and unknown [UNK] token. This resulted in a vocabulary of 4105 tokens. To tokenize an input sequence, the tokenizer started with a class token and then converted the sequence from left to right, matching 6-mer tokens when possible, or using the standalone tokens when necessary (for instance, when the letter N was present or if the sequence length was not a multiple of 6). Similar to our previous work, the first step of the pre-training involved splitting each reference genome into chunks of 6100 nucleotides, where each chunk shared the first and last 50 nucleotides with the preceding and proceeding chunk, respectively. As a data augmentation exercise, for each epoch and chunk, a starting nucleotide index was randomly sampled between 0 and 100, and the sequence was then tokenized from this nucleotide. At each step, a batch of sequences randomly sampled from the epoch set was fed to the model.

### Pre-training strategy
The MLM objective was used to pre-train our model in a self-supervised manner. In a self-supervised learning setting annotations (supervision) for each sequence are not needed as we can mask some proportion of the sequence and use the information contained in the unmasked portion of the sequence to predict the masked locations. This allows us to leverage the vast amount of unlabeled genomic sequencing data available. Specifically, 15% of the tokens in the input sequence are selected to be augmented with 80% being replaced with a mask token, 10% randomly replaced by another token from the vocabulary, and the final 10% maintaining the same token. The tokenized sequence is passed through the model and a cross entropy loss is computed for the masked tokens. Updates to the model parameters are

computed through gradient descent with Adam, the adaptive learning rate optimizer. Pre-training was carried out with a sequence length of 1025 tokens and an effective batch size of 1.5M tokens for 315k update steps, resulting in the model training on a total of 472.5B tokens. The Adam optimizer was used with a modified square decay learning rate (lr) schedule where a warm up period with linear increase was used for 64K steps with initial lr of 0.00005 and end lr of 0.0001. A square decay function was then applied to the lr for all subsequent steps. The exponential decay rates $\beta_1 = 0.9$ and $\beta_2 = 0.999$ were used in the optimizer. See Supplementary Table 1 for training hyper parameter details.

## Fine-tuning strategy

After pre-training our model with the MLM objective we further fine-tune the model in a supervised learning setting using several annotated, plant genomics datasets. We fine-tune our pre-trained model with IA[3][30], a method which injects re-scaling weight vectors at the model's activation's, allowing us to achieve high levels of performance while only computing gradient updates for approximately 1% of the model parameters. The tasks upon which we fine-tune include single output regression, multi output regression, binary classification, multi-label classification.

## Hardware

Model pre-training was carried out using Google TPU-V4 accelerators, specifically a TPU v4-1024 machine containing 512 devices. Since the model was able to fit on a single TPU device, pre-training was carried out in a data parallel fashion where model parameters were replicated and the effective batch sharded across the 512 devices with subsequent gradient accumulation. We trained for a total of approx. four days.

## Polyadenylation site prediction

We obtained data for alternative polyadenylation (APA) sites from PlantAPAdb, which currently represents the most comprehensive manually curated catalog of APA sites in several plants. The dataset is available at http://www.bmibig.cn/plantAPAdb. For our APA site prediction, we focused on five specific species: *Oryza sativa L.* (japonica and indica), *Arabidopsis thaliana, Chlamydomonas reinhardtii, Medicago truncatula,* and *Trifolium pratense.* We downloaded all possible sequences around polyadenylation (poly[A]) sites for each of these species. These sequences encompassed different genomic regions such as 3′ UTR, 5′ UTR, CDS, intron, and exon. Each sequence had a length of 400 bp, with the upstream 300 bp and downstream 100 bp sequence relative to the respective poly(A) site cluster (PAC). The poly(A) site itself was located at the 301st position. To obtain a negative set, we randomly shifted the position of the poly(A) site by both negative and positive values between 1 and 50. We then extracted a 400 bp sequence based on these shuffled positions. The number of negative sequences was twice that of the positive sequences. It is important to note that this strategy of generating negative sequences is more challenging compared to approaches used in other studies. This is because the negative sequences still contain similar gene-related sequences, including those from 3′UTR, 5′UTR, CDS, intron, and exon regions. Furthermore, we emphasize that the model was trained to predict whether a given sequence had a poly(A) site at the 301st position or not. To prevent data leakage, we limited the test set, for each plant species, to sequences that were exclusively derived from a single sampled chromosome. Due to the imbalance in our dataset resulting from the strategy of recreating negative sequences, we evaluated the performance of the models using both the area under the receiver operating characteristic curve (AUROC) and the precision-recall curve (AUPRC). Additionally, we recreated the dataset utilized in Gao et al.[33] for APA site prediction. The positive sequences were sampled from the 16K dataset originally introduced in Loke et al.[86], which consists of over 16,000 downstream sequences from the 3′UTR of *Arabidopsis thaliana.* Following the approach in Gao et al., we initially selected sequences longer than 162 base pairs and then trimmed them to a sequence length of 162 base pairs. These trimmed sequences included 131 base pairs upstream and 31 base pairs downstream of the annotated poly(A) site. To create negative samples,

we randomly sampled 9,704 coding sequences, 3222 intronic sequences, and 501 5′-UTR sequences from the Arabidopsis Information Resources (TAIR) database. This sampling ensured a balanced dataset, as described in Gao et al. Similarly to the positive sequences, we trimmed the negative sequences to a length of 162 base pairs. Since the Gao et al. study did not specify the training and test splitting ratio, we used a standard 0.9/0.1 split ratio, maintaining the same proportion of negative coding, intronic, and 5′-UTR sequences in both the training and test datasets.

## Splicing site prediction

We used the dataset compiled in Baten et al.[87] for splice site prediction, which is available at https://public.bmi.inf.ethz.ch/user/behr/splicing/. The dataset consists of sequences of 398 bp each, containing both acceptor and donor sites, as well as non-acceptor and non-donor sequences across the genome in *Arabidopsis thaliana.* To prevent data leakage, the test set was limited to chromosome 5. Given the highly imbalanced nature of the dataset, with 76,871 acceptor sequences and 2,157,898 non-acceptor sequences, as well as 76,659 donor sequences and 3,311,934 non-donor sequences, we evaluated the performance of the models using both the AUROC and AUPRC.

## Long non-coding RNA prediction

We obtained data for long non-coding RNAs (lncRNA) from the Green Non-Coding (GreeNC) database version 2.0[37], which is available at http://greenc.sequentiabiotech.com/. Our lncRNA prediction focused on six specific species: *Glycine max, Manihot esculenta, Solanum lycopersicum, Sorghum bicolor, Triticum aestivum,* and *Zea mays.* For each of these species, we selected lncRNA sequences with a length smaller than 6,000 bp. Additionally, we only included lncRNA sequences that were reported on scaffolds that were present in the version of the associated annotation file for the given reference assembly mentioned in the GreeNC database. To construct the set of negative sequences, we extracted all mRNA sequences from the reference genome, using the assembly mentioned in the GreeNC database. We removed mRNA sequences longer than 6,000 bp to maintain consistency with the lncRNA dataset. To prevent the model from relying solely on sequence length for classification and to encourage the learning of more meaningful features, we performed a matching process between lncRNA and mRNA sequences based on both sequence length and GC content. Specifically, during the matching process, mRNA sequences were selected if they had a length equal to or less than 100 bp compared to the corresponding lncRNA sequence. Similarly, the GC content of the mRNA sequences was within a range of 1% less or 1% more than the GC content of the corresponding lncRNA sequence. To prevent data leakage, we created a test set for each plant species that exclusively consisted of sequences derived from a single sampled chromosome. In the case of *Triticum aestivum,* the test set was limited to chromosomes 1A, 1B, and 1D. We evaluated the performance of the models using both the AUROC and AUPRC. To assess the models robustness, we also independently resampled 5 times matches numbers of mRNAs used as negative sequences, for each species separately. In addition, we replicated the dataset compiled in Meng et al.[38] to compare our lncRNA prediction performance with that of PlncRNA-HDeep. This dataset was also based on the GreeNC database, but included sequences exclusively from *Zea mays.* This dataset consisted of 18,110 lncRNA sequences from the GreeNC database that were downsampled to generate a positive set of 18,000 samples. The negative set was based on 57,776 mRNA sequences obtained from the RefSeq database, and also downsampled to create a balanced dataset. To ensure a fair comparison, we used the same training and test datasets available at https://github.com/kangzhai/PlncRNA-HDeep/. The only difference was that we trimmed down sequences longer than 6000 bp since our model does not handle longer sequences. This trimming affected 43 sequences in the training dataset and 9 sequences in the test dataset, which are unlikely to greatly affect the performance comparison between models.

## Promoter and terminator strength prediction

For promoter strength prediction, we used the publicly available self-transcribing active regulatory region sequencing (STARR-seq) assays for *Arabidopsis thaliana*, *Zea mays* and *Sorghum bicolor*[40]. The STARR-seq assays included all known promoter regions for the species, which were defined as -165 to +5 bp relative to annotated transcription start sites (TSSs), and their strength was measured in two systems: tobacco leaves and maize protoplasts. To enable direct comparison with the CNN model trained in the original publication, we downloaded their training and test datasets available https://github.com/tobjores/Synthetic-Promoter-Designs-Enabled-by-a-Comprehensive-Analysis-of-Plant-Core-Promoters/, which included the 170 bp promoter sequences and their measured STARR-seq strength in the tobacco leaf and maize protoplast systems. For terminator strength prediction, we used data compiled from a second STARR-seq-based study that measured the activity of over 50,000 terminators from *Arabidopsis thaliana* and *Zea mays* in[41]. These were defined as a 170 nucleotide sequence from position -150 to +20 relative to a cleavage and polyadenylation site. As for the previous study, and to enable a comparison with the CNN model trained in the original publication, we downloaded the training and test datasets available at https://github.com/lampoona/Terminators-Plant-STARR-seq. As in both these studies, we used the coefficient of determination ($R^2$) to evaluate the performance of promoter and terminator strength prediction.

## Chromatin profiles prediction

We used the dataset compiled in Zhao et al.[11] for chromatin profiles prediction. The compiled dataset is based on transposase-accessible chromatin with sequencing (ATAC-seq) data from multiple tissues of six plant species including *Arabidopsis thaliana*, *Oryza sativa*, *Zea mays*, *Setaria italica*, *Sorghum bicolor*, and *Brachypodium distachyon*. NCBI and SRA accession number of the resources and publications used to compile the dataset can be found in the original publication. For data generation, we followed the same strategy as outlined in the publication. We generated the same number of training sequences samples, each of size 1000 bp, retrieved from a reference genome. For each ATAC-seq sample, we labeled each training sequence as 1 (i.e. positive sample) if the middle 200 bp region overlaps with an open chromatin region (OCR) by more than 50% of the sequence length or as 0 (i.e. negative sample) otherwise. To ensure a fair comparison with the Zhao et al. predictions we selected the same chromosomes used as validation and test sets for each plant species. As in the Zhao et al. study, we evaluated the performance of the models using the area under the AUROC and AUPRC metrics.

## Tissue-specific quantitative gene expression level prediction

To test our performance on predicting tissue-specific gene expression levels, we collected data from large-scale studies of expression in five plants (Supplementary Data 2). All the data were downloaded as raw sequencing files and reprocessed with a uniform pipeline. We downloaded all publicly available RNA-seq experiments from the *Arabidopsis thaliana* tissue atlas[88] stored in ArrayExpress[89] (https://www.ebi.ac.uk/biostudies/arrayexpress/studies/E-MTAB-7978), which included 56 experiments across 27 tissues. For *Zea mays*, we downloaded the RNA-seq data published by Walley et al.[90] stored in the GEO Database[91] (https://www.ncbi.nlm.nih.gov/geo/query/acc.cgi?acc=GSE50191). The *Zea mays* dataset consisted of 68 RNA-seq experiments across 23 tissues. For *Oryza sativa*, we collected samples from two NCBI BioProjects (https://www.ncbi.nlm.nih.gov/bioproject/?term=PRJEB47919 and https://www.ncbi.nlm.nih.gov/bioproject/?term=PRJEB32629), totaling 20 experiments across seven tissues. For *Solanum lycopersicum*, we downloaded the RNA-seq experiments associated to the whole-genome sequencing project[92] and stored in the Gene Expression Atlas[93] (https://www.ebi.ac.uk/gxa/experiments/E-MTAB-4812) which included ten tissues with one replicate each. Finally, we also retrieved the *Glycine max* tissue atlas[94] from the European Nucleotide Archive (https://www.ebi.ac.uk/ena/browser/view/SRA012188) which included 14 samples across nine tissues. The genome and annotation version used for gene expression analyses were downloaded from Ensembl Plants version 56[85] and are listed in

Supplementary Data 3. RNA-seq read mapping was performed with the STAR aligner[95] version 2.7.10b. The genome indices for each species were prepared using STAR's *–runMode genomeGenerate* mode and the options *–sjdbOverhang 99 –genomeSAindexNbases 12* using the genome files and gtf genome annotation files listed in Supplementary Data 3. The raw RNA-seq reads were mapped using STAR's *–quantMode GeneCounts* mode, and with the options *–outFilterType BySJout –outFilterMultimapNmax 40 –alignSJoverhangMin 8 –alignSJDBoverhangMin 1 –outFilterMismatchN max 999 –outFilterMismatchNoverReadLmax 0.04 –alignIntronMin 20 –alignIntronMax 1000000*. Within each species, the read counts across all tissues and replicates were imported into R and normalized with DESeq2[96] version 1.38.3. Size factors were first estimated using the *estimateSizeFactors()* method and then used to normalize the reads using the *counts()* method with the *normalized=TRUE* option. For species that had multiple replicates per tissue, we averaged the resulting counts in each tissue to get one value per tissue. Finally, a pseudocount of one was added to each count and they were log2 transformed. For training the model, we followed a similar strategy as outlined in Washburn et al.[51]. We used promoter-proximal sequences based on the TSS annotations and ensured that the sequence overlapped with the coding sequence by extracting 5000 bp upstream and 1000 bp downstream of the TSS, resulting in a sequence of 6000 bp. To prevent potential data leakage, we divided the promoter sequences into gene families using the Washburn et al. pipeline available at https://bitbucket.org/bucklerlab/p_strength_prediction/. Specifically, we conducted an all-by-all BLAST on a given plant's proteome sequence to evaluate pairwise similarity between proteins. As a given gene can encode multiple protein isoforms, the results of the BLAST search were collapsed to gene-level similarity. Then, a graph was constructed with nodes representing genes and edges connecting paralogous genes. The graph was further divided into clusters (i.e., gene families) using a clustering algorithm, and genes not assigned to a family were considered as a family of their own. Using this set of partly independent gene sequences based on their family assignments, we allocated promoter sequences into training and test datasets, thus ensuring that similar promoter sequences were not leaked. Since the number of promoter sequences allocated to families might differ across species, we randomly chose a number of families that would result in an 80%/20% split for each plant species separately. For *Zea mays*, we also examined the percentage of genes in our test set that belonged to a PFAM family present in the training set. This analysis revealed that only 430 genes out of the total 4483 genes in the test set (i.e., less than 10%) were part of such PFAM families, indicating that the approach employed here generated gene families similar to those found in PFAM. Lastly, for all species, we used the coefficient of determination ($R^2$) to evaluate the performance of gene expression prediction.

To further evaluate the performance of AgroNT in predicting gene expression and compare it to a previously developed method, we trained a model specifically for predicting expressed and non-expressed genes. We used the dataset compiled by Washburn et al.[51], which is based on 422 individual samples from seven studies, providing a comprehensive collection of *Zea mays* tissues at various developmental stages. The NCBI and SRA accession numbers can be found in the original publication. The dataset included 39,470 sequences with gene expression labels mentioned in the paper, but sequences were available for only 35,402 of those sequences. As we did not retrieve the missing sequences for training, the comparison to their reported performance becomes somewhat more challenging, as the training dataset was reduced. Following the methodology described by Washburn et al., we trained two separate models, each trained on either promoter or terminator sequences with a fixed size of 1500 bp. We classified each sequence as associated with an expressed gene or a non-expressed gene. As done in the Washburn et al. study, we evaluated the performance of the models using the area under the receiver operating characteristic curve (AUC).

## Token importance for gene expression level prediction

We also evaluated the significance of a genomic region within the 6000 bp promoter-proximal sequence by performing an importance

analysis similar to the one described in[29]. To do this, we randomly replaced each of the 1000 possible tokens within the promoter-proximal sequence with a random token, ensuring that the newly generated token was not the same as the original one. Then, we compared the predicted expression levels of the sequence with the randomly altered token to the predicted expression levels based on the original sequence. In other words, for a given sequence, we generated 1000 new sequences where only a single token differed from the original sequence. We carried out this importance analysis using 100 randomly selected promoter-proximal sequences, drawn only from the test sets, for each of the four plant species that were employed to train the gene expression prediction models. Given that the four plant species gene expression models included various tissues, we averaged the differences across tissues and sequences to derive our final importance scores.

### Zero-shot scores for variant consequence prediction

To calculate zero-shot scores for a specific site of interest, we followed these steps: First, for each single nucleotide polymorphism (SNP), we retrieved a 6000 bp sequence centered on the SNP of interest based on the reference genome of a given species. Next, we created two sequences: one carrying the reference allele and the other carrying the alternative allele at the SNP position. Then, we computed several zero-shot scores that capture different aspects of the vector distances in the embedding space between these two sequences. These scores include the L1 distance (Manhattan), L2 distance (Euclidean), cosine similarity, and dot-product (not normalized cosine similarity). Additionally, we computed a loss-based score, namely the log-likelihood ratio (LLR), which compares the probabilities of alternate and reference alleles.

For the evaluation of zero-shot scores on potential deleterious variants in *Arabidopsis thaliana*, we used the genomes from the 1001 Arabidopsis Genome Project (accessible at https://1001genomes.org/). For *Oryza sativa*, we used the recently compiled database RiceVarMap2[49], which included over 4,000 individual samples from multiple populations (accessible at http://ricevarmap.ncpgr.cn/). In this analysis, we only considered bi-allelic SNPs and removed any SNP with a genotyping rate lower than 95% and 99% in *Arabidopsis thaliana* and *Oryza sativa*, respectively, to ensure reliable allele frequencies. We used a less stringent rate in *Arabidopsis thaliana* as using a threshold of 99% resulted in only 286,672 SNPs. We evaluated a total of 6,494,574 SNPs for *Arabidopsis thaliana* and 3,128,064 SNPs for *Oryza sativa* in these analyses. We note that the number of SNPs in *Oryza sativa* represent a 50% sample of the more than 6 million SNPs that remained after the genotyping rate filter. To annotate missense and non-synonymous SNPs, we used the General Feature Format (GFF) files from the TAIR10 and IRGSP-1.0 reference genomes obtained from Ensembl Plants. For the assessment of zero-shot scores on phenotype-associated variants, which was only conducted in *Arabidopsis thaliana*, we referred to the Ara-GWAS catalog found at https://aragwas.1001genomes.org. Specifically, we considered all variants annotated as "top association". Similar to the previous analysis, we focused on biallelic SNPs. To distinguish between coding and non-coding associated variants, we used the gene coordinates from the GFF of the TAIR10 reference genome. Variants that overlapped with gene coordinates were labeled as coding, while those that did not were labeled as non-coding. The LLR score, referred to as the Genomic Pre-trained Network (GPN) score in[28], as well as the phastCons and phyloP conservation scores, were also included as additional scores for comparison. The GPN scores were computed as described in[28]. Specifically, for each SNP, we retrieved a 512 bp sequence centered on the SNP, which corresponds to the context limit of the GPN model, to estimate the loss of the alternate and reference alleles. To assess the reliability of our computed GPN scores, we checked the correlation of the GPN scores that we computed for *Arabidopsis thaliana* with those provided by the authors at https://huggingface.co/datasets/gonzalobenegas/processed-data-arabidopsis/. We found a strong correlation (r=0.97) between our GPN scores and the pre-computed GPN scores, indicating that the GPN

scores computed for *Oryza sativa* accurately represent the scores intended by the authors. The less than perfect correlation between our computed and the pre-computed GPN values in *Arabidopsis thaliana* is likely due to the fact that in the study, the loss of the alternate and reference alleles was based on the average of the forward and reverse strands, a step we omitted to speed up computation. The pre-computed phastCons and phyloP conservation scores were downloaded from PlantRegMap (accessible at https://plantregmap.gao-lab.org/). As the two conservation scores relied on multiple alignments across species to compute a score, resulting in some SNPs lacking annotations, we ensured fair comparison across methods by considering only variants with associated scores across all methods.

### In silico mutagenesis in the cassava genome

To leverage the capability of AgroNT for obtaining accurate predictions across a variety of regulatory features, we further trained AgroNT to predict enhancer elements and gene expression across multiple tissues in the understudied crop, *Manihot esculenta* (cassava). For enhancer elements prediction, we used the PRO-seq data generated in *Manihot esculenta* seedlings from the study conducted by Lozano et al.[97]. Specifically, we used a set of 9665 intergenic regulatory elements (IRE) regions, which the authors defined as those located at least 1000 bp away from any gene. To maintain consistency and avoid reprocessing the PRO-seq data, we used the *Manihot esculenta* reference genome v6.1 from Phytozome, which corresponds to the reference genome employed in the study. The training dataset was constructed as follows: For positive sequences, which represent positive enhancer samples, we extracted a 1000bp sequence centered in the middle of the PRO-seq peak. To create matched negative sequences for each positive sequence, we randomly selected a 1000bp intergenic sequence from the same chromosome, ensuring it neither overlapped with any PRO-seq peak nor differed significantly in GC% from the corresponding positive sequence. Specifically, the GC% of the negative sequence was within 5% points of the GC% of the positive sequence. This approach resulted in a balanced and GC-matched dataset. We then split the training and test sets by chromosomes, ensuring they were strictly non-overlapping. Chromosomes 17 was designated as the test set, and chromosome 9 as the validation set. The remaining chromosomes were used as the training set. The model's performance was evaluated using the AUROC metric.

For gene expression prediction, we used the gene expression atlas from the study conducted by Wilson et al.[98], which measured gene expression levels across 11 tissue/organ types. We employed the reference genome and annotation from Ensembl Plants version 56 and followed the same pipeline to process the RNA sequencing data described above. As for our previous gene expression models, we used promoter-proximal sequences based on TSS annotations to obtain sequences of 6000 bp and followed the same clustering approach to assign each promoter sequence to a gene family. We then used this information to produce a training and test set, using an 80%/20% split. Based on these trained models, we performed a comprehensive profiling of regulatory potential. Specifically, for a given sequence obtained from the reference genome, we systematically mutated each site to all three possible distinct nucleotides. In our initial analyses to detect motifs, we predicted the impact of all possible single-nucleotide substitutions across 200 bp upstream and 200 bp downstream of the midpoint of the top 1000 best predicted enhancer regions. Similarly, for the gene expression analyses, we conducted all possible single-nucleotide substitutions across promoter sequences located within 200 bp upstream and 200 bp downstream of the TSS. We selected 1000 promoter sequences of genes that had the highest average gene expression levels across all tissues. To measure the impact of a mutation, we computed the log-fold change as $LFC = log_2(P_1/P_0)$. Here, $P_0$ represents the probability (or gene expression level) predicted for the original sequence, and $P_1$ represents the probability (or gene expression level) predicted for the mutated sequence. Given the observation that AgroNT has gained an understanding of the importance of disrupting critical motifs in enhancer and promoter-proximal regions concerning gene expression

values, we also assessed all possible single-nucleotide substitutions within 50 bp upstream and 50 bp downstream of the midpoint for all enhancer regions. Similarly, we performed all possible single-nucleotide substitutions within 50 bp upstream and 50 bp downstream of the TSS for all promoter-proximal regions. This resulted in a total of ~ 2.9 and ~ 7.3 million mutations, respectively, which we have made available for future research purposes.

We visualized the mutation scores of a given sequence using the ggseqlogo function from the R package ggseqlogo. To identify transcription factor motifs corresponding to patterns predicted to be important for enhancer or promoter activity, we mapped the positions of 656 plant TF motifs from the non-redundant Jaspar CORE 2022 collection (https://jaspar.genereg.net/download/data/2022/CORE) in these sequences. We accomplished this using the matchMotifs function from the R package motifmatchr, with the following parameters: p.cutoff = 1e-04 and bg = "even". The figure panel highlights TF motif types that match positions with high prediction mutation scores.

## Statistics and reproducibility

For the long non-coding RNA prediction analyses, we used a Wilcoxon rank sum test in the R package stats (version 4.2.1) to assess the difference in length between mRNA and long non-coding RNA sequences. For the promoter and terminator strength analyses, when comparing the observed and predicted values, we estimated the coefficient of determination ($R^2$) using the lm function in the R package stats (version 4.2.1). For the chromatin accesibility analyses, we used a Wilcoxon rank sum test in the R package stats (version 4.2.1) to assess the difference in performance of AgroNT with a model based on the DeepSEA architecture. For the gene expression analyses, when comparing the observed and predicted values, we estimated the coefficient of determination ($R^2$) using the lm function in the R package stats (version 4.2.1). For the zero-shot based prediction analyses, the error bars around the point estimates were based on a sampling with replacement strategy using the sample function in the R package base (version 4.2.1). For the correlation between zero-shot scores the Pearson's product moment correlation coefficient was computed using the cor.test function in the R package stats (version 4.2.1). The area under the receiver operating characteristic (ROC) curve (AUC) and the area under the precision-recall curve (AUPRC) for all analyses related to a classification task were computed using the roc.curve and pr.curve functions in the R package PRROC (version 1.3.1). Data manipulation and processing analyses were performed using the dplyr (version 1.1.1), tidyr (version 1.3.0), purrr (version 1.0.1), tibble (version 3.2.1), and stringr (version 1.5.0) packages in R. The graphs in Figs. 2–8 were created using the ggplot2 (version 3.4.3) package in R. R version 4.2.1 was used throughout all analyses. Additional data manipulation analyses were conducted using the packages Pandas (version 1.5.3) and NumPy (version 1.23.5) in Python (version 3.10.9).

## Data availability

Pre-training reference sequences were obtained from publicly available resources and are listed in Supplementary Data 1 and can also be found via HuggingFace at https://huggingface.co/datasets/InstaDeepAI/plant-multi-species-genomes. Gene annotations were obtained from Ensembl Plants (https://plants.ensembl.org) and Phytozome (https://phytozome-next.jgi.doe.gov/). A comprehensive description of the datasets, including links for all the raw datasets presented in the study, can be found in the Methods section. The Plant Genomic Benchmark and the processed data that support the findings of this study have been made publicly available via HuggingFace at https://huggingface.co/datasets/InstaDeepAI/plant-genomic-benchmark (https://doi.org/10.57967/hf/2464).

## Code availability

The pre-trained AgroNT model has been made publicly available via HuggingFace at https://huggingface.co/InstaDeepAI/agro-nucleotide-transformer-1b (https://doi.org/10.57967/hf/2465). It can also be accessed via the following public GitHub repository: https://github.com/instadeepai/nucleotide-transformer. The code for pre-training the model has not been made available as it is proprietary. All additional code has been made available in the associated HuggingFace and GitHub repositories.

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

## Acknowledgements
This research was supported by Cloud TPUs from Google's TPU Research Cloud (TRC). We also extend our gratitude to the researchers who deposited experimental data in public databases, those who maintain these databases, and those who make analytical and predictive methods accessible to the scientific community.

## Author contributions
TP and ML conceived the research idea. JM-R, ET, LG, MR, HD-T, BDA, and GR performed the analyses. JC, NLC, MS, AL, KB, TP, and ML provided advice on study design and analyses. JM-R, ET, BDA, TP, and ML wrote the paper with input from all co-authors.

## Competing interests
The authors declare no competing interests.
