## [Peer Review File · Communications Biology]

Reviewers' comments:

Reviewer #1 (Remarks to the Author):

This manuscript presents a study on a plant multi-species genomics large-scale language model. While there are still gaps compared to similar studies in humans/animals in terms of innovative ideas, model design, downstream applications, and case analyses (e.g., 10.1038/s41586-023-06139-9 and 10.1101/2023.01.11.523679), the efforts made in this research are commendable. As the first study of its kind in plant multi-species genomics, this manuscript integrates a set of test datasets for reliable evaluation of downstream applications of the model and demonstrates the superior performance of fine-tuned pre-trained models on a series of downstream tasks. Taking the cassava genome as an example, this paper provides a case study and resources for exploring causal variation through virtual saturation mutagenesis. It is foreseeable that this paper will contribute to the advancement of crop genomics large-scale language model research. In summary, this manuscript addresses a popular and important topic and makes relevant contributions to the field.

Some Comments:

1. Both of the two main research outcomes mentioned in this manuscript, the pre-trained model and benchmark dataset (<https://huggingface.co/InstaDeepAI/agro-nt> and <https://huggingface.co/datasets/InstaDeepAI/plant-genomic-benchmark>), are inaccessible. I have confirmed that during the same time, <https://huggingface.co/InstaDeepAI> is accessible, but the PGB resources described in the text cannot be found at <https://huggingface.co/datasets?sort=trending&search=InstaDeepAI>. Therefore, I am unable to determine the availability and ease of use of the pre-trained model and integrated data resources described in this paper, which are crucial factors in assessing the contributions of this research.
2. The provided case study on saturated mutagenesis in the cassava genome in this manuscript lacks solid evidence to support the claim that the model can assist in discovering causal regulatory variations. We noticed that the authors referenced PlantDeepSEA in this paper, which presents two cases, "Discovering high-impact sites within the maize QTL UPA2" and "Prioritizing non-coding causal polymorphisms in the rice gene DEP1," that can be reproduced by the DeepSEA model. We are interested to know if this study can reproduce these two cases through in silico mutagenesis or if it can discover other reported cases.
3. Using an 80%/20% split for gene expression prediction is not reasonable since genes within the same gene family often exhibit similar expression patterns. Therefore, it is important to ensure that the genes in the test set come from different gene families than those in the training set. Here, we provide a plant gene family database for reference: <https://bis.zju.edu.cn/cropgf/>
4. To construct a balanced dataset, this paper mentions downsampling of negative samples multiple times. Regarding this, it would be helpful if the authors could provide more detailed information, such as the sampling random seed, to demonstrate that the negative sample sampling is sufficiently random. If feasible, conducting multiple tests with different sets of negative samples would enhance the credibility of the model evaluation results.
5. The authors mention that "we propose the use of the multiple datasets encompassing seven distinct genomic prediction tasks, which have been compiled here, as the Plants Genomic Benchmark (PGB)." However, the PGB contains a limited amount of data, which is sufficient as test cases for this study but falls short of a comprehensive benchmark dataset for the field. The authors may consider modifying their description or attempting to create a larger dataset to establish a solid foundation for testing methods in this field (which we eagerly anticipate but is not mandatory). Here, we provide some plant databases and datasets for reference: Databases: Expression (<https://biotec.njau.edu.cn/plantExp/>), Methylation (<https://ngdc.cnbc.ac.cn/methbank/>), circRNAs (<http://ibi.zju.edu.cn/plantcircbase/>), Chromatin Accessibility (<https://bioinfor.nefu.edu.cn/PlantCADB/>), Post-translational Modifications (<http://qptmplants.omicsbio.info/>), lncRNAs (<http://justrna.itps.ncku.edu.tw/>). Datasets: TSS (<https://doi.org/10.1111/tpj.15957>), Terminator Strength (<https://doi.org/10.1101/2023.06.16.545379>).
6. The RiceVarMap v2.0 database provides variant effect scores obtained using the Basenji model trained on rice chromatin accessibility data. SoyMD, on the other hand, provides variant effect scores

obtained using the Basenji2 model trained on soybean multi-omics data. Both of these databases serve as good comparison benchmarks to illustrate the performance of downstream variant effect annotation tasks using the pre-trained models discussed in this paper. The authors may consider discussing these databases in relation to their work.

7. Can the specific number of samples be labeled on Figure 1a?

8. For Figure 2a and 2b, it would be better to avoid line breaks between the English term for rice and the specific cultivar names.

9. For the comparative analysis of the regression models, the presentation method used in this study (e.g., Figure 3a) is not clear. It is suggested to consider using a scatter plot instead, with the accuracy of predictions (correlation between true and predicted values) for different models on the x-axis and y-axis. Each point on the plot represents a different sample from the test set.

Reviewer #2 (Remarks to the Author):

Mendoza-Revilla et al. in "A Foundational Large Language Model for Edible Plant Genomes" propose a large language model (LLM), called AgroNT, which is trained on the genomes of 48 different agronomic plant species. The LLM is pre-trained in a first self-supervised phase to reconstruct masked DNA sequences of size 6000bp. DNA sequences are tokenized using non-overlapping 6-mers. They then used the pre-trained LMM embeddings to train and fine-tune several different classification and regression-based models for seven different genomic prediction tasks. The paper is very well written with an impressive collection of different applications of AgroNT. The results are promising and show, similar to other publications, that pre-trained LLMs can be very beneficial in increasing the predictive power of various biological sequence-based problems. I do not have much to criticize in the main paper and the applications of AgroNT. However, I have a rather strong opinion about the methods part. The paper lacks basic details on how the model was built, trained, and evaluated. There are not enough details in the paper to reproduce the results nor to reproduce the architecture of the model. I miss details on hyperparameter tuning and model evaluation. The link to the pre-trained model on Hugging Face does not work. It was not possible to evaluate and try out the pre-trained AgroNT model. In terms of open science, I advocate publishing not only the pre-trained model, but also the code on how these models were trained and tuned. Otherwise it is not possible to verify the results. And this is very important when working with genetic and genomic data. Having a pre-trained LMM for 48 plant genomes is nice to have and a great opportunity for the community as a resource. But not knowing how the model was trained nor being able to reproduce the results makes the model less interesting to use in one's own applications.

Title: Response to Reviewer Comments

We are grateful for the thorough review and insightful comments provided on our manuscript. The constructive feedback has been invaluable in refining our work. In the following pages, you will find a detailed response addressing each of the comments raised by both reviewers. We have tried to address each point to the best of our ability. We hope that our responses prove satisfactory to your questions and comments and contribute to the enhancement of our study. Thank you once again for your time and consideration.

Reviewers' comments:

Reviewer #1 (Remarks to the Author):

This manuscript presents a study on a plant multi-species genomics large-scale language model. While there are still gaps compared to similar studies in humans/animals in terms of innovative ideas, model design, downstream applications, and case analyses (e.g., 10.1038/s41586-023-06139-9 and 10.1101/2023.01.11.523679), the efforts made in this research are commendable. As the first study of its kind in plant multi-species genomics, this manuscript integrates a set of test datasets for reliable evaluation of downstream applications of the model and demonstrates the superior performance of fine-tuned pre-trained models on a series of downstream tasks. Taking the cassava genome as an example, this paper provides a case study and resources for exploring causal variation through virtual saturation mutagenesis. It is foreseeable that this paper will contribute to the advancement of crop genomics large-scale language model research. In summary, this manuscript addresses a popular and important topic and makes relevant contributions to the field.

Response: *We are grateful to the reviewer for dedicating time to evaluate our manuscript and for recognizing the potential contribution of our work to the progress of large-scale language model research in crop genomics.*

Some Comments:

1. Both of the two main research outcomes mentioned in this manuscript, the pre-trained model and benchmark dataset (<https://huggingface.co/InstaDeepAI/agro-nt> and <https://huggingface.co/datasets/InstaDeepAI/plant-genomic-benchmark>), are inaccessible. I have confirmed that during the same time, <https://huggingface.co/InstaDeepAI> is accessible, but the PGB resources described in the text cannot be found at <https://huggingface.co/datasets?sort=trending&search=InstaDeepAI>. Therefore, I am unable to determine the availability and ease of use of the pre-trained model and integrated data resources described in this paper, which are crucial factors in assessing the contributions of this research.

Response: *We apologize for not providing external users access to both the model and the benchmarking downstream tasks. We have now made the model accessible via the following*

link: <https://huggingface.co/InstaDeepAI/agro-nucleotide-transformer-1b> and the downstream tasks can be accessed through this link: <https://huggingface.co/datasets/InstaDeepAI/plant-genomic-benchmark>. Model descriptions have been included, and the updated links are provided in the revised manuscript. We have now modified the text in our manuscript with the correct links in the Abstract, Code, and Data availability sections.

We also want to highlight that both our model and downstream tasks datasets on Huggingface are compatible with the previous Nucleotide Transformer models and datasets. As such the Nucleotide Transformer notebooks provided to fine-tune the models will work here if one simply changes the model and tasks' names. Example notebooks to fine-tune the models can be accessed through the links:

https://github.com/huggingface/notebooks/blob/main/examples/nucleotide_transformer_dna_sequence_modelling.ipynb and https://github.com/huggingface/notebooks/blob/main/examples/nucleotide_transformer_dna_sequence_modelling_with_peft.ipynb.

2. The provided case study on saturated mutagenesis in the cassava genome in this manuscript lacks solid evidence to support the claim that the model can assist in discovering causal regulatory variations. We noticed that the authors referenced PlantDeepSEA in this paper, which presents two cases, "Discovering high-impact sites within the maize QTL UPA2" and "Prioritizing non-coding causal polymorphisms in the rice gene DEP1," that can be reproduced by the DeepSEA model. We are interested to know if this study can reproduce these two cases through in silico mutagenesis or if it can discover other reported cases.

Response: *Following the reviewer's suggestion we have now evaluated whether our chromatin accessibility models can also retrieve the strongest signals in one of the genes, namely DEP1, that was evaluated in the PlantDeepSEA study. Encouragingly, the strongest effects were obtained for the vg0916410299 variant, one of the 9 non-coding variants shown previously to have a direct effect on DEP1, which was also the variant highlighted by plantDeepSEA. This suggests that our models can indeed be used to prioritize phenotype-associated variants. We have expanded the results section to include this new result (Results subsection: Chromatin accessibility prediction).*

3. Using an 80%/20% split for gene expression prediction is not reasonable since genes within the same gene family often exhibit similar expression patterns. Therefore, it is important to ensure that the genes in the test set come from different gene families than those in the training set. Here, we provide a plant gene family database for reference: <https://bis.zju.edu.cn/cropgf/>

Response: *We agree with the reviewer that randomly splitting the dataset for the gene expression prediction task is not reasonable. As described in the Methods section (subsection A.6.6) we divided the promoter sequences into gene families using the Washburn et al. pipeline and used these families to assign promoter sequences into training splits. As shown in the mentioned publication, this strategy prevents potential data leakage. We apologize if this was*

not made clear in our manuscript and have edited the associated section to make this more clear. In addition, we also evaluated how many genes present in our test set were from a pfam family (based on the database that the reviewer pointed out [cropgf]) that was also present in our training genes. Out of all the 4483 genes present in our test set only 430 were assigned to a pfam family that was also present in the training gene set (i.e. <10%), suggesting that our approach produced similar gene families as those present in pfam. We have also included this additional analysis in the relevant section (Methods Subsection: Tissue-specific quantitative gene expression level) for clarity.

Reference:

Washburn, J. D., Mejia-Guerra, M. K., Ramstein, G., Kremling, K. A., Valluru, R., Buckler, E. S., & Wang, H. (2019). Evolutionarily informed deep learning methods for predicting relative transcript abundance from DNA sequence. *Proceedings of the National Academy of Sciences*, 116(12), 5542-5549.

4. To construct a balanced dataset, this paper mentions downsampling of negative samples multiple times. Regarding this, it would be helpful if the authors could provide more detailed information, such as the sampling random seed, to demonstrate that the negative sample sampling is sufficiently random. If feasible, conducting multiple tests with different sets of negative samples would enhance the credibility of the model evaluation results.

Response: *In response to the reviewer's suggestion, we have conducted additional analyses to further validate the credibility of our model evaluation. Specifically, we performed 5 random downsamples of the negative observations in the long non-coding RNA dataset across all presented downstream tasks. Encouragingly, the model exhibited consistent performance across these random downsamples, with minimal variation (median range of AUROC values from 1.6 to 3.9 points, and AUPRC values from 2.5 to 4.6 across species). This result suggests that the model demonstrates high robustness. We have incorporated these new analyses into the pertinent section of the manuscript (Results Subsection: Polyadenylation site, splice site, and long non-coding RNA prediction).*

5. The authors mention that "we propose the use of the multiple datasets encompassing seven distinct genomic prediction tasks, which have been compiled here, as the Plants Genomic Benchmark (PGB)." However, the PGB contains a limited amount of data, which is sufficient as test cases for this study but falls short of a comprehensive benchmark dataset for the field. The authors may consider modifying their description or attempting to create a larger dataset to establish a solid foundation for testing methods in this field (which we eagerly anticipate but is not mandatory). Here, we provide some plant databases and datasets for reference: Databases: Expression (<https://biotec.njau.edu.cn/plantExp/>), Methylation (<https://ngdc.cncb.ac.cn/methbank/>), circRNAs (<http://ibi.zju.edu.cn/plantcircbase/>), Chromatin Accessibility (<https://bioinform.nefu.edu.cn/PlantCADB/>), Post-translational Modifications (<http://qptmplants.omicsbio.info/>), lncRNAs (<http://justrna.itps.ncku.edu.tw/>). Datasets: TSS (<https://doi.org/10.1111/tpj.15957>), Terminator Strength (<https://doi.org/10.1101/2023.06.16.545379>).

Response: *We thank the reviewer for suggesting additional downstream tasks to further improve the comprehensiveness of our proposed Plants Genomics Benchmark (PGB). Upon investigating each of the studies and resources we have decided to include the “Terminator strength” downstream task mainly because (1) it represents an additional regression-based task, which tends to be more challenging than classification-based ones, (2) is not a downstream task that is already present in PGB, and (3) the associated publication included the training and test datasets, including their associated predictions based on a CNN model, which would allow us to compare the performance of our models to that of the authors. We show that AgroNT either surpasses or matches the performance of the CNN-based model in 5 out of 6 datasets, and only slightly underperforms in one. We have detailed the inclusion of this additional downstream task in the methods section and present our model performance in the results section of the manuscript (Results subsection: Promoter and terminator activity prediction).*

6. The RiceVarMap v2.0 database provides variant effect scores obtained using the Basenji model trained on rice chromatin accessibility data. SoyMD, on the other hand, provides variant effect scores obtained using the Basenji2 model trained on soybean multi-omics data. Both of these databases serve as good comparison benchmarks to illustrate the performance of downstream variant effect annotation tasks using the pre-trained models discussed in this paper. The authors may consider discussing these databases in relation to their work.

Response: *In response to the reviewer's suggestions, we have incorporated a new analysis utilizing the RiceVarMap v2.0 database. While we did not directly compare our chromatin accessibility models to those of RiceVarMap due to potential differences in experimental assays, we did compare our models to DeepSEA, which serves as a robust baseline for this task. Instead, we leveraged the extensive collection of over 4,000 individual rice samples to evaluate the performance of zero-shot scores in detecting deleterious variants (defined as <0.001 and 0.001-0.01 frequency). For this purpose, we used the recently developed GPN model as a zero-shot score baseline, which was previously applied solely in our analysis on Arabidopsis thaliana. We consider this an enhancement to our zero-shot analysis since we had previously evaluated its utility only on a single species, Arabidopsis thaliana, whereas for all other tasks, we had included several species. To reflect these updates, we have expanded the Methods and Results sections accordingly to provide comprehensive documentation of our approach and findings (Results subsection: Zero-shot prediction of functional variants).*

7. Can the specific number of samples be labeled on Figure 1a?

Response: *We have now included the specific number of reference genomes in Figure 1a.*

8. For Figure 2a and 2b, it would be better to avoid line breaks between the English term for rice and the specific cultivar names.

Response: *Following the reviewer's suggestion, we have now removed the line breaks in the panels.*

9. For the comparative analysis of the regression models, the presentation method used in this study (e.g., Figure 3a) is not clear. It is suggested to consider using a scatter plot instead, with the accuracy of predictions (correlation between true and predicted values) for different models on the x-axis and y-axis. Each point on the plot represents a different sample from the test set.

Response: *We apologize if the figure legend was not clear enough. Figure 3a was presented not as an approach to illustrate the performance of the model but rather to depict the distribution of the promoter strength values that we aimed to predict. The scatterplots mentioned by the reviewer are displayed in Figure 3b. We have slightly modified the figure legend text to emphasize that the values shown in Figure 3a are derived from the molecular assay and not our model.*

Reviewer #2 (Remarks to the Author):

Mendoza-Revilla et al. in "A Foundational Large Language Model for Edible Plant Genomes" propose a large language model (LLM), called AgroNT, which is trained on the genomes of 48 different agronomic plant species. The LLM is pre-trained in a first self-supervised phase to reconstruct masked DNA sequences of size 6000bp. DNA sequences are tokenized using non-overlapping 6-mers. They then used the pre-trained LMM embeddings to train and fine-tune several different classification and regression-based models for seven different genomic prediction tasks. The paper is very well written with an impressive collection of different applications of AgroNT. The results are promising and show, similar to other publications, that pre-trained LLMs can be very beneficial in increasing the predictive power of various biological sequence-based problems.

Response: *We appreciate the reviewer's time spent evaluating our manuscript and acknowledging the potential benefits of our pre-trained LLM in enhancing the predictive capabilities of diverse biological sequence-based challenges.*

I do not have much to criticize in the main paper and the applications of AgroNT. However, I have a rather strong opinion about the methods part. The paper lacks basic details on how the model was built, trained, and evaluated. There are not enough details in the paper to reproduce the results nor to reproduce the architecture of the model. I miss details on hyperparameter tuning and model evaluation. The link to the pre-trained model on Hugging Face does not work. It was not possible to evaluate and try out the pre-trained AgroNT model. In terms of open science, I advocate publishing not only the pre-trained model, but also the code on how these models were trained and tuned. Otherwise it is not possible to verify the results. And this is very important when working with genetic and genomic data. Having a pre-trained LLM for 48 plant genomes is nice to have and a great opportunity for the community as a resource. But not knowing how the model was trained nor being able to reproduce the results makes the model less interesting to use in one's own applications.

Response:

Unfortunately, we are unable to share the proprietary code used for pre-training, as it has been optimized specifically for our computing infrastructure. However, we share the reviewer's commitment to transparency and reproducibility. In line with this, we have taken several steps to facilitate reproducibility of our results.

Firstly, we have made the weights and inference code of our model available on Hugging Face (<https://huggingface.co/InstaDeepAI/agro-nucleotide-transformer-1b>), allowing others to utilize the model for inference purposes. Additionally, we have provided the datasets for all downstream tasks on the same platform (<https://huggingface.co/datasets/InstaDeepAI/plant-genomic-benchmark>), ensuring accessibility to the data used in our experiments.

Furthermore, both the model and datasets are compatible with the previous nucleotide transformer format. This means that the notebooks used for fine-tuning the nucleotide transformer (such as those found at https://github.com/huggingface/notebooks/blob/main/examples/nucleotide_transformer_dna_sequence_modelling_with_peft.ipynb and https://github.com/huggingface/notebooks/blob/main/examples/nucleotide_transformer_dna_sequence_modelling.ipynb) can be adapted to reproduce all fine-tuning experiments simply by changing the model and dataset names.

Additionally, we have provided the pre-training dataset we used on Hugging Face (<https://huggingface.co/datasets/InstaDeepAI/plant-multi-species-genomes>). While we do not share the pre-training code, we have adapted our model and datasets to adhere to Hugging Face standards. Therefore, users can attempt to replicate our pre-training experiment using Hugging Face support (see https://huggingface.co/docs/transformers/main/tasks/masked_language_modeling). However, we caution users that training times may be longer than those reported, and the associated compute costs may be prohibitive compared to the relatively low cost of fine-tuning.

Finally, in response to the reviewer's request, we have enhanced the methods section to provide all experimental details and hyperparameters, ensuring that users have the necessary information to reproduce our experiments under the same conditions see new Supplementary Fig. 1, Supplementary Table 1 and Methods subsections Architecture and Pre-training).

REVIEWERS' COMMENTS:

Reviewer #1 (Remarks to the Author):

The author has satisfactorily addressed the concerns I previously raised, and I firmly believe that this study can make a substantial contribution to the progress of plant genome large language models. Consequently, I am of the opinion that this manuscript is suitable for acceptance and publication.

Reviewer #2 (Remarks to the Author):

The authors addressed all my concerns and made the code (at least the pre-trained models and the inference code) and data available to the scientific community. Although the authors cannot provide the code for pre-training the model due to proprietary issues, the authors provide links to replicate some of the pre-training experiments from the paper. I recommend publication of the paper.

Title: Response to Reviewer Comments

Reviewer #1 (Remarks to the Author):

The author has satisfactorily addressed the concerns I previously raised, and I firmly believe that this study can make a substantial contribution to the progress of plant genome large language models. Consequently, I am of the opinion that this manuscript is suitable for acceptance and publication.

Response: *We thank the reviewer for confirming that all comments have been addressed. We are grateful to the reviewer for dedicating time to evaluate our manuscript and for recognizing the contribution of our work to the progress of large-scale language model research in crop genomics.*

Reviewer #2 (Remarks to the Author):

The authors addressed all my concerns and made the code (at least the pre-trained models and the inference code) and data available to the scientific community. Although the authors cannot provide the code for pre-training the model due to proprietary issues, the authors provide links to replicate some of the pre-training experiments from the paper. I recommend publication of the paper.

Response: *We express gratitude to the reviewer for verifying that all comments have been addressed and for their understanding regarding the unshareable proprietary code related to this work. We would also like to emphasize our appreciation to the reviewer for dedicating time to evaluate our manuscript.*